# Operando dynamics of trapped carriers in perovskite solar cells observed via infrared optical activation spectroscopy

Jiaxin Pan[1], Ziming Chen[1] ✉, Tiankai Zhang[2], Beier Hu[1], Haoqing Ning[1], Zhu Meng[1], Ziyu Su[1], Davide Nodari[1], Weidong Xu[1], Ganghong Min[1], Mengyun Chen[2], Xianjie Liu[3], Nicola Gasparini[1], Saif A. Haque[1], Piers R. F. Barnes[4], Feng Gao[2] & Artem A. Bakulin[1]

Conventional spectroscopies are not sufficiently selective to comprehensively understand the behaviour of trapped carriers in perovskite solar cells, particularly under their working conditions. Here we use infrared optical activation spectroscopy (i.e., pump-push-photocurrent), to observe the properties and real-time dynamics of trapped carriers within operando perovskite solar cells. We compare behaviour differences of trapped holes in pristine and surface-passivated $FA_{0.99}Cs_{0.01}PbI_3$ devices using a combination of quasi-steady-state and nanosecond time-resolved pump-push-photocurrent, as well as kinetic and drift-diffusion models. We find a two-step trap-filling process: the rapid filling (~10 ns) of low-density traps in the bulk of perovskite, followed by the slower filling (~100 ns) of high-density traps at the perovskite/hole transport material interface. Surface passivation by n-octylammonium iodide dramatically reduces the number of trap states (~50 times), improving the device performance substantially. Moreover, the activation energy (~280 meV) of the dominant hole traps remains similar with and without surface passivation.

Remarkable optoelectronic properties and low manufacturing costs make metal halide perovskites promising materials for next-generation photovoltaic techniques. Substantial progress in improving perovskite solar cell (PeSC) performance has been achieved in the last decade, bringing the certified power conversion efficiency (PCE) to 26.1% and making PeSCs the fastest-developing solar cell technology in the history of photovoltaics[1,2]. Currently, the performance of PeSCs is approaching the Shockley–Queisser limit of a single cell and their further development requires the fundamental reduction of non-radiative recombination in PeSC devices[3]. Electronic defects, known as carrier 'traps', are largely responsible for non-radiative recombination under one-sun excitation[4,5]. Therefore, deeper understanding of the nature of trap states and the dynamic behaviour of the carriers being trapped in PeSCs is

critical to minimise the voltage loss to further improve the device performance.

The origin of trap states in perovskite has been extensively studied in the past decade[6,7]. Point defects (e.g., interstitials, substitutions, and vacancies), line defects, grain boundaries in perovskite lattice, as well as impurities are considered the main sources of trap states[8,9]. Fortunately, most structural defects in perovskites induce the formation of shallow electronic trap states within or close to the edge of valence band (VB) or conduction band, rather than deep electronic trap states[10,11]. This provides perovskites with excellent defect tolerance[12]. Carrier transport and recombination are only modestly affected by traps at room temperature as thermal energy is sufficient to release a substantial number of trapped carriers back to band states prior to recombination, minimising the negative effect of

[1]Department of Chemistry and Centre for Processible Electronics, Imperial College London, London W12 0BZ, UK. [2]Department of Physics, Chemistry and Biology (IFM), Linköping University, Linköping SE-58183, Sweden. [3]Laboratory of Organic Electronics, ITN, Linköping University, Norrköping SE-60174, Sweden. [4]Department of Physics, Imperial College London, London SW7 2AZ, UK. ✉e-mail: z.chen@imperial.ac.uk

defect formation[11]. However, many carriers still become immobilised, particularly at interfacial states and eventually contribute to trap-assisted non-radiative recombination, decreasing the open-circuit ($V_{oc}$) as well as limiting short-circuit current ($J_{sc}$) and fill factor (FF) in a device[7,13–15]. Reducing the overall trap density in perovskites is therefore essential for optimising device performance.

Multiple strategies have been suggested to reduce the density and the influence of traps in perovskites, including film formation optimisation, crystal strain relaxation, compositional engineering, crystallisation control, etc.[16–19]. Among them, passivating traps at the interfaces between perovskite layer and charge extraction layers is a popular and effective route to boost the $V_{oc}$ of devices[20,21]. Utilising Lewis base-Lewis acid interaction between perovskites and passivating molecules as well as introducing large organic cations to form a passivation layer [or even two-dimensional/three-dimensional (2D/3D) hybrid perovskites] on top of the perovskite surface are two main strategies to improve PeSC performance[22–26]. Generally, the effect of surface passivation is simply attributed to a significant reduction of the surface trap density relative to the unmodified perovskite crystal surface[27]. However, besides the number of trap states, their energy, cross-section, and location also influence the device[28–30]. Moreover, the dynamics of carrier interactions with the trap states (e.g., the time scale for carriers to fill, release and recombine from trap states) are also critical for understanding the impact of traps on PeSC performance and for further device optimisation.

Conventional optical spectroscopic techniques struggle to selectively observe the dynamics of carrier trapping in solar cells. For example, in transient absorption spectroscopy (TAS), the spectrum of trapped carriers significantly overlaps that of free carriers which generally also have a much larger population than the population in traps so that any signal from trapped carriers is swamped by the free carrier signal. In addition, it is difficult to directly monitor carriers in trap states with photoluminescence (PL) based methods due to the non-emissive nature of recombination via these states. Instead, non-radiative trap-assisted recombination can often be deduced indirectly from a PL signal by establishing simple radiative models[31]. However, the resulting picture can be unclear with various possible scenarios for trapped carrier dynamics. Finally, TAS and PL methods are difficult to apply to working photovoltaic devices under typical solar illumination conditions.

Thermal energy or infrared (IR) light can be used to activate or probe the trapped carriers to yield trap information. For example, Kobbekaduwa et al. developed a temperature-dependent ultrafast photocurrent spectroscopy technique to evaluate the depth of shallow traps and estimate the timescale of thermally induced de-trapping. In the ultrathin active layers they studied, trapping dynamics occurred on a picosecond time scale although the dynamics of recombination were not detectable with the technique[32]. Alternatively, El-Zohry1 et al. recently used visible pump-IR probe measurements to monitor the population of trapped carriers in perovskite films within several ns after photo-excitation. Although not directly relevant to device optimisation, these measurements provided substantial fundamental insight and suggested a qualitative mode of carrier trapping dynamics[33]. Time-resolved optical pump-terahertz probe spectroscopy has been applied to detect the formation of small polarons ('self trapping') from photogenerated charge carriers since the terahertz signal is proportional to the conductivity of delocalised carriers and hence sensitive to the loss in conductivity as they from small polarons[34]. However, this method measures the average mobility of carriers and may not be sensitive to small fractions of holes or electrons trapped in defect states. In our previous work, a femtosecond optical pump-IR push-photocurrent (PPPc) technique, which is specifically sensitive to trapped carriers, was used to observe the ultra-fast trap-filling process in a PbS solar cell. Unfortunately, the limited time window prohibited the assessment of slow trap-filling and trap-assisted recombination processes[35].

Herein, we introduce ns−ms IR optical activation PPPc spectroscopies to overcome the limitations of existing spectroscopic techniques, and to reveal the dynamics of trapped carriers in working PeSCs. By taking advantage of photocurrent detection, PPPc can successfully overcome the limitations of conventional optical methods[36]. These methodologies also have the advantage of allowing trapped carriers to be monitored selectively and sensitively with relatively low optical-excitation intensities or fluences. In these techniques, band-edge carriers are generated by an optical "pump" beam, following/during this some of the carriers become trapped. Then the sub-bandgap energy photons of an IR "push" beam are absorbed by a fraction of the trapped carriers to excite them back to their band states. The IR de-trapped carriers (which might otherwise have recombined) then contribute to an additional photocurrent generated by the device. Therefore, the amplitude of IR-induced current can be used to evaluate the concentration of trapped carriers in the device. The experiment can be done by using the beams of continuous-wave (CW) lasers with slow (µs−ms) modulation to simply assess and compare the extent of charge trapping in different devices. Alternatively, short fs−ns pulses can be used as pump and push to evaluate the dynamics of trap filling and the recombination kinetics of trapped carriers.

We investigated the behaviour of trapped carriers in pristine and surface-passivated $FA_{0.99}Cs_{0.01}PbI_3$ [where FA = $CH(NH_2)_2$] PeSCs, as well as established corresponding kinetic and drift-diffusion models for the trapped carrier dynamics. And we found that there were two clear contributions to the trapping processes coming from the traps in the bulk of perovskite and traps at the interface with the charge (hole) extraction layer. We also observed that interface passivation substantially reduced the total number of trapping sites in the device but did not alter the energy of traps. To the best of our knowledge, this is the first report completely and directly revealing the dynamics of trap filling and trap-assisted recombination processes in working PeSCs.

## Results

To control the density of traps and evaluate the effects of defect-reduction strategies in PeSCs, we fabricated $FA_{0.99}Cs_{0.01}PbI_3$ PeSCs with and without surface passivation. The architecture of the pristine (non-passivated) device was indium tin oxide (ITO)/$SnO_2$/$FA_{0.99}Cs_{0.01}PbI_3$/2,2',7,7'-Tetrakis[N,N-di(4-methoxyphenyl)amino]−9,9'-spirobifluorene (Spiro-OMeTAD)/Au, while that of the surface-passivated device was ITO/$SnO_2$/$FA_{0.99}Cs_{0.01}PbI_3$/n-octylammonium iodide (OAI)/Spiro-OMeTAD/Au (Fig. 1a). OAI, which is widely used with perovskites, served to passivate interfacial traps between the $FA_{0.99}Cs_{0.01}PbI_3$ and the spiro-OMeTAD[37,38]. These device compositions and architectures were selected for their excellent ambient (-months), light, and thermal stability without the need for encapsulation as well as for being a typical model system studied across the perovskite photovoltaic community. According to the device performance shown in Fig. 1b and Supplementary Table 1, compared with the pristine device, the surface-passivated device had a larger $V_{oc}$, which suggests the trap-assisted recombination at the perovskite surface was effectively suppressed. The improved PL quantum yield of the surface-passivated perovskite film (13.1%), compared with that of the pristine sample (4.9%), further confirms the successful passivation of interfacial traps by OAI molecules (Supplementary Fig. 1a). This can be attributed to the simultaneous passivation of $FA^+$ and $I^-$ interface vacancies by the OAI molecules, as shown in Fig. 1c and discussed in Supplementary Fig. 1b–d. The low-angle (2θ < 10°) X-ray diffraction signal shown in Supplementary Fig. 1e, f shows no indication of a 2D (or layered) perovskite phase formed on top of the 3D perovskite surface after the deposition of OAI.

Figure 2a shows the concept of the experimental setup for quasi-steady-state PPPc spectroscopy. The setup has a very simple layout of

 

two diode lasers (pump and IR push), beam-overlap optics, a chopper, a lock-in amplifier, and a device holder. The pump and push beams are overlapped in space and focused on the sample. The chopper modulates the IR push beam and sends the reference signal to the lock-in amplifier to detect the IR-induced photocurrent ($\triangle J_{IR}$). The chopper

can also be moved to the pump beam to detect the pump-induced photocurrent ($J_{Pump}$). CW-based PPPc setup (CW-PPPc) uses an 808-nm CW laser as pump beam and a 980-nm CW laser as push beam with chopper modulation in a 37−4000 Hz range, which allows us to access the information of trapped carriers in µs to ms time scale. In addition,

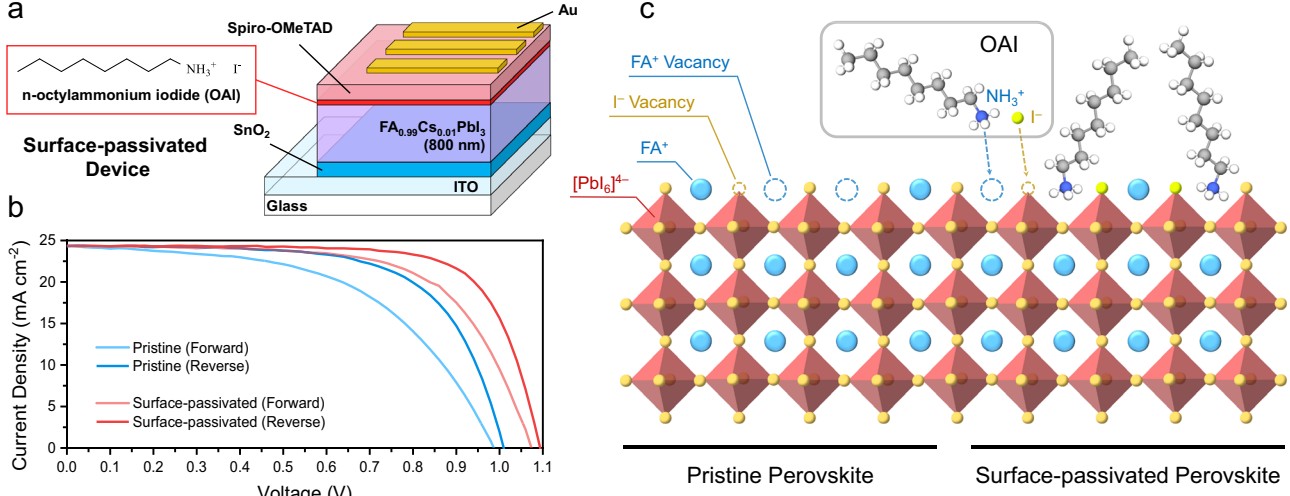

**Fig. 1 | Device properties of FA$_{0.99}$Cs$_{0.01}$PbI$_3$ PeSCs. a** Device architecture of the surface-passivated device. The pristine device has a similar architecture but without the OAI layer. **b** Current density versus voltage of the pristine and surface-passivated devices measured during forward and reverse voltage sweeps at a scan rate of 40 mV s$^{-1}$. **c** Configuration of the surfaces of pristine and surface-passivated perovskites. The OA chains mount to the perovskite surface via their NH$_3^+$ groups, which also passivate the FA$^+$ vacancies simultaneously, while the I$^-$ vacancies can be passivated by the I$^-$ in OAI[64].

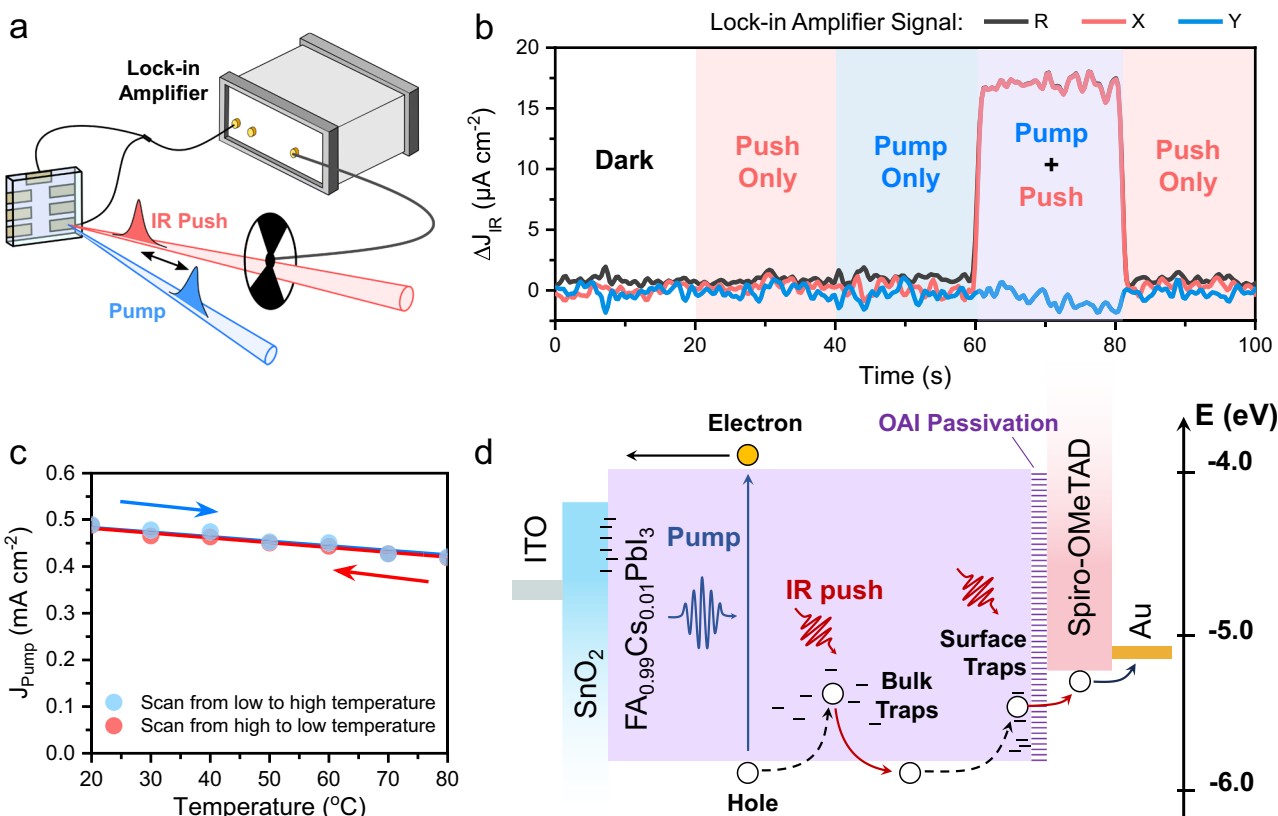

**Fig. 2 | Experimental setup and working mechanism of PPPc. a** Schematic diagram of CW- and ns-PPPc setups. **b** $\triangle J_{IR}$ responses of FA$_{0.99}$Cs$_{0.01}$PbI$_3$ device under different illumination conditions in CW-PPPc measurement. The $\triangle J_{IR}$ only responded when both pump and push beams illuminated the device simultaneously. **c** FA$_{0.99}$Cs$_{0.01}$PbI$_3$ device photocurrent ($J_{Pump}$) as a function of temperature from 20 °C to 80 °C and from 80 °C to 20 °C at a constant pump intensity. **d** Flat-band energy diagram of the studied devices and the working mechanism of PPPc showing pathways of hole dynamics.

ns-resolution PPPc (ns-PPPc) uses an 800-nm pulse laser (with ~40 fs pulse duration) as pump beam and a 1064-nm pulse laser (with ~0.5 ns time resolution) as push beam, which allows us to access transient information for trapped carriers from ns to μs time scales. The timing between the pump and push is controlled by an electrical delay generator triggering push pulses with an accuracy of ~10 ns (due to triggering jitter). The selection of pump and push wavelengths depends on the absorption profile of $FA_{0.99}Cs_{0.01}PbI_3$, as shown in Supplementary Fig. 2.

Figure 2b shows CW-PPPc for $FA_{0.99}Cs_{0.01}PbI_3$ device under different illumination. When the sample is not illuminated or illuminated by just pump or push beam, the $R$ signal of lock-in amplifier (i.e., total IR-induced current) is negligible, because the steady-state current from pump ($J_{Pump}$) is not picked up by the lock-in amplifier and the IR push photons alone do not directly generate detectable photocarriers. When the $FA_{0.99}Cs_{0.01}PbI_3$ device is illuminated by both pump and push beams simultaneously, the $\triangle J_{IR}$ signal soars up, which suggests that $\triangle J_{IR}$ originates from the intragap states populated by pump-generated mobile carriers. By comparing $X$ (in-phase) and $Y$ (out-of-phase) signal from the lock-in amplifier in Fig. 2b and Supplementary Fig. 3, we conclude that there is no significant change of phase at various chopper modulation frequencies. The amplitude of $\triangle J_{IR}$ scales linearly (Supplementary Fig. 4) with the intensity of IR light, indicating that $\triangle J_{IR}$ can be used as a measure of trapped carrier concentration[39]. This linear relationship also suggests that only a small fraction of trapped carriers are depopulated by IR photons.

Note that the IR push beam can de-trap carriers via two possible mechanisms: (i) via direct optical transition from trap to the band or (ii) via thermal activation, after the energy from absorbed IR photons increases the temperature of the film. To investigate the possible de-trapping process caused by the IR heating effect, we evaluated the response of the sample to changes in temperature. Figure 2c shows that upon heating, the photocurrent decreases, suggesting that the increase of the $\triangle J_{IR}$ signal does not originate from an IR heating effect. Moreover, we estimate that any change in photocurrent induced by IR-heating is at least two orders smaller than the PPPc signal under the same illumination conditions (Supplementary Note 1), further indicating that IR heating has a negligible impact on our results.

In this case, the $\triangle J_{IR}$ signal should solely originate from the optical transition between the filled trap states and band states, as shown in Fig. 2d which depicts the process for holes (a similar process is expected for the electron route). After the generation by the pump beam, band-edge free holes drift/diffuse to the anode, sampling local environment with occasional capture by trap states in the material bulk and at the interfaces. The trapped holes are immobile, and many will recombine from these states and thus not contribute to the photocurrent. The IR push photons have a chance to optically re-activate these immobile holes from the trap states back to VB before recombination, and hence generate additional photocurrent detected as $\triangle J_{IR}$. We note that band-edge free carriers can also be excited by IR push beam to higher energy states, however, this would cause no significant change in photocurrent because i) exciting free carriers to hot states does not change the number available for extraction from the device, and ii) hot carrier trapping is typically not observed in 3D perovskites[40,41]. Therefore, PPPc spectroscopy is a highly sensitive and selective technique for monitoring carriers trapped in states that act as recombination centres in the device (further discussion of the origin of PPPc signal can be found in Supplementary Note 2). Through analysing $\triangle J_{IR}$ signal, we can obtain information about the trapped carrier population from ns to ms time scales, via the combination of CW- and ns-PPPc spectroscopies.

## Trapped carrier concentration and density of traps

We varied the intensity of pump and push beams in CW-PPPc measurement to investigate how the trapped carrier concentration and the densities of traps were different in the pristine and surface-passivated devices. Figure 3a and Supplementary Fig. 5 show the results of the intensity dependence experiments for the surface-passivated and pristine $FA_{0.99}Cs_{0.01}PbI_3$ samples, respectively. As $\triangle J_{IR}$ not only relates to the density of traps, but also involves the factor of charge extraction, we therefore present the data in the form of $\triangle J_{IR}/J_{Pump}$ to reflect the ratio between the trapped and free carriers[39]. Both $\triangle J_{IR}$ and $\triangle J_{IR}/J_{Pump}$ vary linearly with IR intensity, indicating that push photons only depopulated a minority of trapped carriers. Also, $J_{Pump}$ signals are very close to linear as a function of pump intensity (Supplementary Fig. 6), indicating that first order (or pseudo-first order) recombination was the dominant loss mechanism in the device under working conditions.

To quantitatively analyse the trap filling behaviour in the device, we calculated the relative concentration of trapped carriers ($n_{TC}$) from a relationship involving the slope of $\triangle J_{IR}/J_{Pump}$ against IR intensity (see Fig. 3a and Supplementary Note 3). Figure 3b shows that pristine device had a much higher $n_{TC}$ than the surface-passivated device at the same pump intensity, suggesting that more band-edge carriers were trapped in the pristine device. Moreover, $n_{TC}$ increased with the increase of pump intensity in both cases, and we found that the $n_{TC}$ and pump intensity followed a power law:

$$n_{TC} \propto I_{pump}{}^{b} \tag{1}$$

The fitted $b$ value reflects whether the trap states are abundant and ready to be filled up ($b = 1$) or are mostly full and close to saturation ($b = 0$). Figure 3b shows that the extracted $b$ values of the pristine and surface-passivated devices were 0.94 and 0.15, respectively. These results illustrate that trap states in the surface-passivated device already tended to be saturated even though the $n_{TC}$ was relatively small, while trap states in the pristine device were still far from saturation although the $n_{TC}$ was large. This phenomenon can be attributed to a dramatic reduction of trap density in the surface-passivated device compared to the pristine. Moreover, considering OAI passivated the top surface of perovskite that collects holes, as well as the significant change of $n_{TC}$ before and after surface passivation, we therefore considered hole traps dominating the total trap states of perovskite.

To cross-check these results and quantitatively estimate the trap density ratio between pristine and surface-passivated devices, we introduced a kinetic model (Supplementary Fig. 7) to analyse the PPPc data. Through globally fitting the CW-PPPc results by such a model, as shown in Supplementary Fig. 8a, b (parameters are summarised in Supplementary Table 2 and details of the fitting can be found in Supplementary Note 4), we successfully extracted the relationship between the modelled trapped carrier concentration and pump intensity (Supplementary Fig. 8c). Its trend agrees with the experimental results shown in Fig. 3b, confirming our interpretation. Notably, the fitted trap density of the surface-passivated device is ~50 times lower compared with that of the pristine device, reflecting that OAI is an effective surface passivator for perovskites.

The dependence of $\triangle J_{IR}/J_{Pump}$ on IR chopper modulation frequency is shown in Fig. 3c. It is apparent that the signal decreases as the frequency increases for both samples, while the phase shift of the signal (Supplementary Fig. 3) remained constant and close to zero over the entire frequency range (37–4000 Hz). A frequency-dependent phase shift would be expected if the variation in the CW-PPPc signal were induced by IR heating, which we did not observe. To determine a characteristic timescale of the process resulting in the frequency-dependent signal, we fitted the data with the Cole-Cole equation commonly used to analyse the chopper frequency-dependent

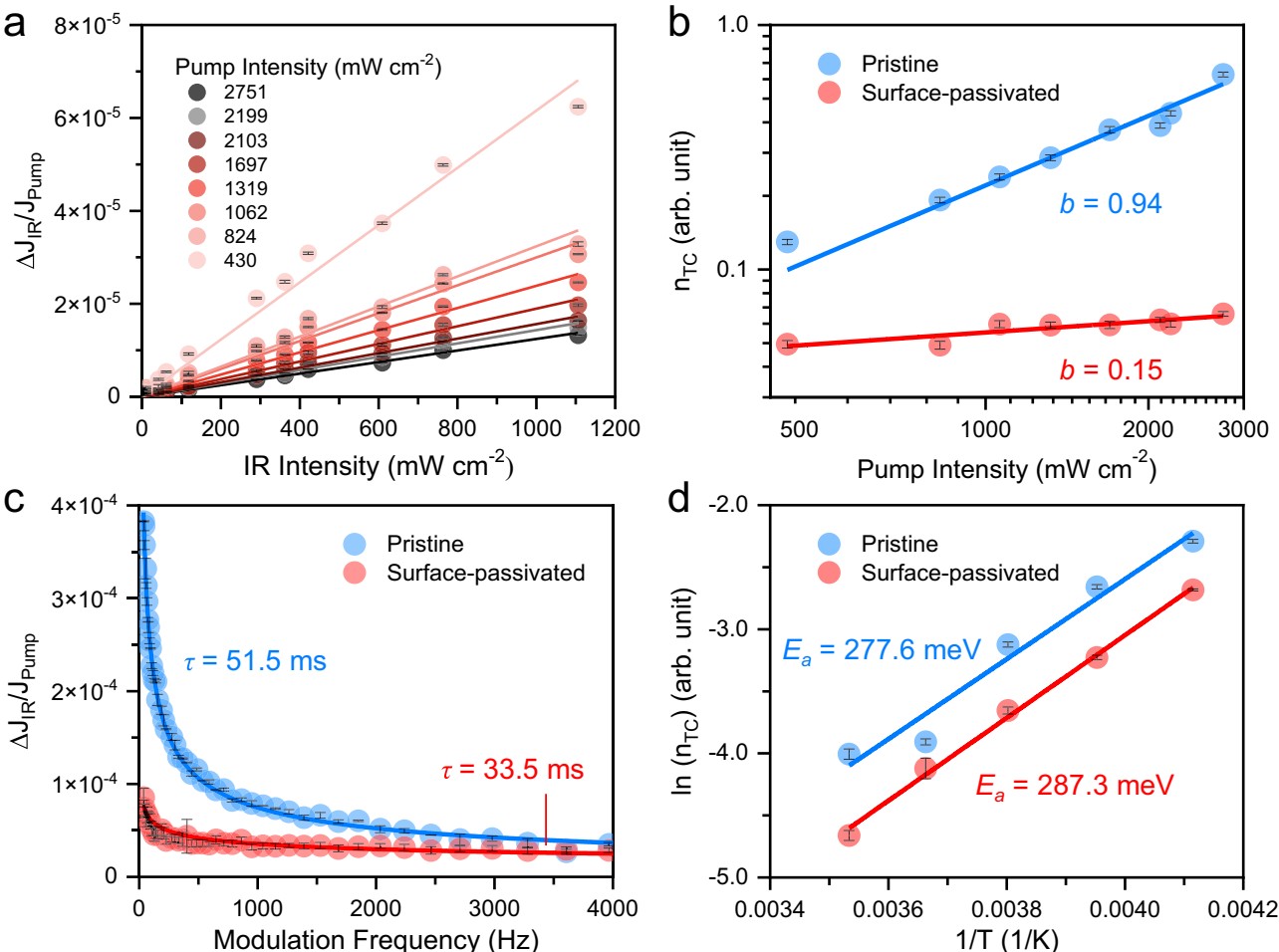

**Fig. 3 | Trapped carrier concentration in pristine and surface-passivated FA$_{0.99}$Cs$_{0.01}$PbI$_3$ devices. a** $\triangle J_{IR}/J_{Pump}$ versus IR push intensity of a surface-passivated FA$_{0.99}$Cs$_{0.01}$PbI$_3$ PeSC. The solid lines indicate the linearly fitted results. **b** Concentration of trapped carriers as a function of pump intensity. The solid lines indicate the fitted results according to Eq. 1. **c** $\triangle J_{IR}/J_{Pump}$ signal as a function of modulation frequency. The solid lines indicate the fitted results according to Eq. 2. **d** Arrhenius plot of the temperature-dependent trapped carrier concentration. The solid lines indicate the fitted results according to Arrhenius equation. The error bars represent the standard deviation of the data.

measurements[42, 43]:

$$\frac{\triangle J_{IR}}{J_{Pump}} = \frac{\left(\frac{\triangle J_{IR}}{J_{Pump}}\right)_0}{1+(i2\pi f \tau)^\alpha} \quad (2)$$

where the $\left(\triangle J_{IR}/J_{Pump}\right)_0$ is the projected steady-state $\triangle J_{IR}/J_{Pump}$ value at 0 Hz; $f$ is the chopper modulation frequency; $\tau$ represents the characteristic time constant for the process and the exponent $\alpha$ reflects the dispersity or range of timescales of the underlying process (if $\alpha = 1$ the process is non-dispersive and the process is increasingly dispersive as $\alpha \to 0$). For the pristine device $\tau = 52$ ms and $\alpha = 0.52$; for the surface passivated device a slightly faster but considerably more dispersive process was observed ($\tau = 34$ ms and $\alpha = 0.28$). These timescales are considerably longer than the series resistor-capacitor (RC) time constant of the devices which put a lower limit on the time for charge extraction (determined from electrochemical impedance spectroscopy to be RC = 0.29 μs and 0.23 μs for pristine and surface-passivated PeSCs, respectively, see Supplementary Fig. 9). We tentatively suggest these highly dispersive time constants reflect the influence of recombination via a subset of traps states with a wide range of depths or capture cross-sections in the device, particularly those at the interfaces. At the interfaces where the majority free carrier species is at a

minimum due to repulsion by accumulated mobile ionic defects, recombination times can be very long. Some of the dispersion may also result from a wide range of IR absorption cross-section for the trapped carriers, some of which may be so weakly absorbing that they will only contribute to $\triangle J_{IR}/J_{Pump}$ when illuminated for a sufficiently long time. The increase in dispersity observed in the passivated device appears to indicate a broader and less modal distribution of trap depths, capture cross-sections (with associated recombination times), or absorption coefficients, remaining after surface passivation. This can be attributed to the less domination of surface traps in the total trap states after the passivation process.

To elucidate the energetics of trap states and to evaluate whether the passivation process has an impact on the characteristic trap depth, we conducted temperature-dependent CW-PPPc measurement. The devices were measured in a nitrogen-filled environment to prevent potential degradation of perovskite under vacuum conditions[44]. Fig. 3d shows that with the decrease of temperature from 300 K to 250 K, the total concentration of trapped carriers increased. This phenomenon was caused by the suppression of thermally de-trapping process at low temperatures, which led to more carriers remaining in the intragap states[45]. Fitting an Arrhenius model to the data, we found the dominant activation energy of traps to be ~280 meV. Both the pristine and surface-passivated devices had similar trap activation energy values, suggesting the surface-trap passivation process

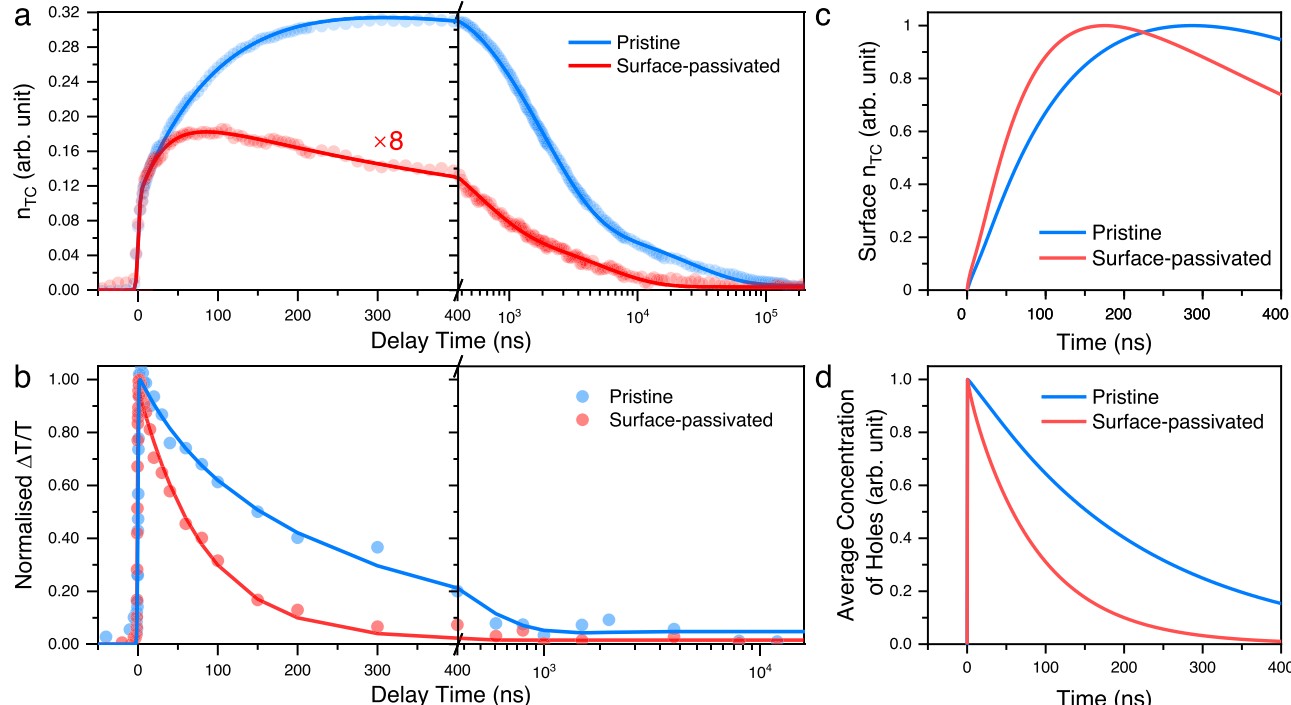

**Fig. 4 | ns-PPPc and ns-TAS results of pristine and surface-passivated devices.** **a** $n_{TC}$ dynamics measured under 5.1 μJ cm$^{-2}$ pump fluence and 170.8 μJ cm$^{-2}$ push fluence. The signal of the surface-passivated device has been multiplied 8 times for better comparison. The rise time of pristine and surface-passivated devices is 142 ns and 33 ns, respectively, which is defined as the time that signals reach 90% of the

maximum value. **b** Normalised ns-TAS dynamics (ground state bleach) measured using a 532-nm pump (0.35 μJ cm$^{-2}$). The decay time (i.e., half-lifetime) of pristine and surface-passivated devices is 156.2 ns and 56.2 ns, respectively. Modelled $n_{TC}$ dynamics at (**c**) perovskite surface and (**d**) average hole concentration in perovskite layer based on the drift-diffusion model (see Supplementary Note 6 for details).

primarily reduced the density of traps and had a minor effect on their characteristic depth.

**Trap-filling dynamics revealed by ns-PPPc**

To reveal the dynamics of trapped carrier population, we performed PPPc experiment using synchronised pulsed laser sources. In agreement with CW-PPPc observations, the devices appeared to be in a first-order recombination regime (Supplementary Fig. 10), and the $R$ value (i.e., total IR-induced current) of the detected signal provides a good description of population dynamics (Supplementary Fig. 11). In such a case, $n_{TC}$ as a function of time can be calculated from the time-resolved $\Delta J_{IR}/J_{Pump}$ data (detailed can be found in Supplementary Note 5). According to Fig. 4a, the maximum amplitude of $n_{TC}$ signal is one order higher in the pristine device than that of the surface-passivated device, which agrees with the CW-PPPc results and indicates very substantial suppression of traps by the OAI interface modifier. The ns-PPPc kinetics show instant growth of $n_{TC}$ in both the pristine and surfaced-passivated devices, which suggests that considerable trap-filling processes occurred within the time resolution of our setup (~10 ns). Such a fast process likely originated from the trap filling by the photocarriers generated in the spatial vicinity of the trap states, and we therefore attribute this process to the filling of bulk traps in both devices[46].

The instant initial growth of $n_{TC}$ is followed by a slower rise in ~100 ns timescale. The relative contribution of slow rise is much higher in the pristine device (~60%) compared to that in the surface-passivated device (~25%). Also, the estimated rise time of $n_{TC}$ in the pristine device (142 ns) was substantially longer than that in the surface-passivated device (33 ns). This indicates that the filling of trap states in the pristine device occurred more slowly, which could be associated with a higher density of trap states, slower carrier diffusion, or presence of different trap sub-ensembles, for instance, 'bulk' and 'surface' traps[47]. The clear bimodal character of $n_{TC}$ rise

(sub-10 ns, plus ~100 ns) supports the latter hypothesis[48]. Consequently, we attribute the instantaneous component to the carrier trapping in the bulk material and the delayed rise to the drift/diffusion-mediated trapping of carriers at the interfaces with charge extraction layer. The fact that passivating the perovskite surface on the hole-extraction side dramatically reduces the delayed trapping supports this hypothesis and indicates again that the majority of the trapped carriers were holes. The change in the delayed PPPc growth timescale in this case could be related to the formation of an positively charged interfacial charge layer due to the accumulated trapped holes at the perovskite surface[49]. These interfacial holes in the pristine sample might screen the internal field, slowing down the drift/diffusion of carriers to the interface[50]. This phenomenon was also observed in a previous report when holes accumulated at the perovskite/spiro-OMeTED interface[51].

To further understand the charge diffusion/drift properties, we conducted ns-based TAS (ns-TAS) which monitored the average carrier concentration in the perovskite layer in ns–μs time scale (Fig. 4b). ns-TAS shows that carrier concentration in surface-passivated perovskite decayed faster (56.2 ns) than that in pristine perovskite (156.2 ns). We attribute these timescales to the carrier extraction, as these timescales agree well with previously reported carrier extraction time based on time-resolved PL and photocurrent measurements[52–55]. The observed timescales also agree with the timescales of the delayed PPPc components, indicating both describe the arrival of photogenerated carriers to the interface between perovskite and hole extraction layer. In addition, the recombination kinetics for trapped carriers are similar in the pristine and surface-passivated samples with only the longest component (~25 μs) being more pronounced in the pristine sample. Furthermore, the timescales of both bulk trap filling and trap-assisted recombination are consistent with those obtained from trapped-carrier modelling based on the previous report[56].

To verify the above scenario, we performed an extended drift-diffusion simulation for charge concentration dynamics at the hole-extraction interface[57, 58]. The realisation of codes is described in Supplementary Note 6 including the device parameters. The extraction rate on the hole transporter side in the pristine device was set smaller than that in the surface-passivated device, mimicking the existence of a stronger positively charged interfacial charge layer in the pristine device. Figure 4c shows the simulated concentration of $n_{TC}$ at the hole extraction interface as a function of time, which qualitatively and quantitatively reproduces experimental ns-PPPc data. The average hole concentration in the active layer shown in Fig. 4d also agrees well with the results of ns-TAS. The successful reproduction of ns-PPPc and ns-TAS data by our drift-diffusion model confirms that interfacial trapping plays a dominant role in both carrier recombination and transport in the device. It also confirms that the drift/diffusion of charges to the interface can be affected by surface passivation.

Furthermore, we verified the results by varying the pump fluence from 0.9 to 5.1 μJ cm$^{-2}$ as shown in Supplementary Figs. 12 and 13. Increased $n_{TC}$ with the increasing pump fluence was observed in both devices, similar to the CW-PPPc results. The normalised $n_{TC}$ dynamics of pristine and surface-passivated devices under various pump and push fluence (Supplementary Figs. 12, 13) appear to be very similar, which highlights the robustness of the ns-PPPc approach. And it also reflects that the trap filling and trap-assisted recombination processes of every single trapped carrier is independent to the others (i.e., the energy exchange between trapped carriers in different trap states is negligible) and no trap healing/annihilation present[59,60]. We consider this might be due to the special isolation among different defects in the perovskite lattice. This finding is also important to understand the behaviour of carriers in trap states.

To investigate how the trapped carrier behaviour changes in higher-performance devices, we optimised the devices via depositing an ultra-thin polymethyl methacrylate (PMMA) layer on top of the SnO$_2$ layer [i.e., with device architectures of ITO/SnO$_2$/PMMA/FA$_{0.99}$Cs$_{0.01}$PbI$_3$/(OAI)/Spiro-OMeTAD/Au]. PMMA can passivate the bottom surface of perovskite via Lewis-base-Lewis-acid interaction between the C = O group and the uncoordinated Pb, resulting in a reduction of overall trap states[55,61,62]. Consequently, much improved PCEs from 16.1% to 19.8% for pristine device and from 19.5% to 23.2% for surface-passivated device were achieved (detailed photovoltaic performance can be found in Supplementary Fig. 14 and Supplementary Table 3). Supplementary Fig. 15a, b show the ns-PPPc results for devices with and without PMMA, illustrating that the overall $n_{TC}$ decreases in both pristine and surface-passivated devices after the incorporation of PMMA. This result further confirms the passivation effect of PMMA. Moreover, in the optimised devices, the rise time of pristine device (92.16 ns) is much longer than that in surface-passivated device (16.86 ns) (Supplementary Fig. 15c), which consolidates the photophysical picture developed for the original pristine and surface-passivated devices.

Using optimised devices as a model system, we conducted temperature-dependent ns-PPPc measurement in 300−240 K region (Supplementary Fig. 16). The main features of the dynamics at room temperature shown in Supplementary Fig. 16a, b stay the same and can be described by the model presented above. The recombination processes substantially slow down, elevating the baseline and making amplitude analysis problematic when high repetition rates are used. We also observed slower rise in both pristine and surface-passivated devices when decreasing the temperature below 260 K, indicating that the reduction of carrier diffusion has a substantial contribution to charge transport at these conditions[4]. An alternative explanation can be the higher positive charge density built up by trapped holes at the perovskite/Spiro-OMeTAD interface due to the reduction of trap-assisted recombination rates and more efficient trapping at lower temperatures.

## Discussion

Herein, we introduced two different PPPc techniques (i.e., CW- and ns-PPPc) to reveal the behaviour of trapped carriers from ns to ms time scale in working PeSCs for the first time. We successfully applied these techniques to investigate the trap-filling and trap-assisted recombination processes in pristine and surface-passivated devices. We found that the filling of bulk traps was fast (within 10 ns) due to the trapping of photocarriers generated in the spatial vicinity of the trap states, while the filling of surface traps was relatively slow as drift/diffusion of band-edge carriers to the perovskite surface was involved in this trapping process. Moreover, we found, the filling of surface trap states leads to the formation of an interfacial charge layer that screens the internal field and hence decelerating the drift/diffusion of carriers to the interface. Besides, our PPPc techniques can also reveal the trapped carrier population change in different devices with different trap densities. A faster saturation behaviour of trapped carrier concentration was found in the device with lower trap density (surface-passivated device), compared with the device with higher trap density (pristine device). Also, a kinetic model was established to estimate the trap density ratio between these two different devices and a ~50 times lower trap density was found after surface passivation. Finally, the nearly identical activation energy of trap state bands in both devices indicates that surface passivation process solely reduces the number of traps and could not change the depth of traps.

Our results successfully demonstrate that PPPc techniques are powerful and highly sensitive to reveal the dynamic, concentration, and activation energy of trapped carriers, facilitating comprehensive understanding of the role of trap states in optoelectronic devices. It provides the field with a new approach to access trap states. We also believe it can be incorporated with other experiments to provide more operando information of traps in a working optoelectronic device, such as the degradation of device under different circumstance.

## Methods

### Precursor preparation

The perovskite FA$_{0.99}$Cs$_{0.01}$PbI$_3$ active layer precursor was prepared by dissolving 1.81 M lead iodide (Anhydro Beads, 99.999% Sigma-Aldrich), 1.65 M formamidium iodide (Greatcell solar), 0.58 M methylamonium chloride (Xi'an Polymer Light Technology Corp.), 0.016 M Cesium iodide (99.999% Sigma-Aldrich) in 1 ml anhydrous N,N-dimethylformamide (99.8%, Sigma-Aldrich) and dimethyl sulfoxide (99.9%, Sigma-Aldrich) mixed solution with the volume ratio of 8:1. The ion-modulated radical doping of spiro-OMeTAD was prepared by mixing 90 mg ml$^{-1}$ spiro-OMeTAD in the chlorobenzene solution with 6 mol% of spiro-OMeTAD$^{2+}$(TFSI$^-$)$_2$ and 18 mol% TBMP$^+$TFSI$^-$. PMMA solution was prepared by dissolving PMMA (Mw = ~4000) in chlorobenzene with a concentration of 2 mg ml$^{-1}$.

### Device fabrication

All ITO substrates were cleaned sequentially in deionized water and ethanol for 15 min respectively, and dried by a compressed nitrogen gun. After 15 min UV-Ozone surface treatment, the SnO$_2$ electron transport layer was deposited by spin coating a 1:6 diluted SnO$_2$ nanoparticle water solution (Alfa Aeser) at 4000 rpm for 30 s, followed by annealing at 150 °C for 20 min in air. For the device with PMMA layer, the PMMA solution was spin-coated (4000 rpm for 30 s) on top of the SnO$_2$ layer and followed by annealing at 100 °C for 8 mins in a N$_2$-filled glovebox. The 800-nm perovskite layer was then deposited via spin coating the perovskite precursor at 5000 rpm for 30 s. Within 10 s of the 5000-rpm spinning, 100 μL of chlorobenzene as the antisolvent was dropped onto the film. After spin-coating, the perovskite film was then annealed at 150 °C for 15 min in ambient air. Then 5 mg ml$^{-1}$ OAI (Greatcell solar) solution in tert-butanol (anhydrous, 99.5%, Sigma-Aldrich) was spin-coated onto the perovskite surface at 5000 rpm and annealed at 100 °C for 3 min for the surface passivation. Later, the hole

transport layers (ion-modulated radical doped spiro-OMeTAD) were deposited by spin coating at 5000 rpm for 30 s without further annealing. A metal electrode (80-nm Au) was finally deposited through thermal evaporation method under a vacuum degree higher than $3 \times 10^{-6}$ Torr to accomplish the solar cell fabrication. A 0.06 cm$^2$ shadow mask was used to define the effective working area of the solar cells. All the devices were used unencapsulated.

## Perovskite film and device characterisation

Ultraviolet-visible absorption spectra were measured with a PerkinElmer model Lambda 900. For PL quantum yield measurement, the perovskite films were directly deposited on glass substrates. The excitation source was a 635-nm CW laser and the excitation power was 100 mW. The signal was detected by the spectrometer (andor spectrometer, Oxford Instruments) through an integrated sphere. XPS measurement (monochromatised Al K$\alpha$ $h\nu$ = 1486.6 eV) was carried out using a Scienta-200 hemispherical analyser with dedicated home designed and built spectrometer. Data were calibrated by referencing to Fermi level and Au $4f_{7/2}$ peak position of the Ar$^+$ ion sputter-clean gold foil. The experimental condition was set so that the full width at half maximum of the clean Au $4f_{7/2}$ line (at the binding energy of 84.00 eV) was 0.65 eV. X-ray diffraction (XRD) patterns were obtained from an X-ray diffractometer (Panalytical X'Pert Pro) with an X-ray tube (Cu K$\alpha$, $\lambda$ = 1.5406 Å). Photocurrent–voltage curves were measured (2400 Series Source Meter, Keithley Instruments) in a N$_2$-filled glovebox and under simulated AM 1.5 sunlight at 100 mW cm$^{-2}$ irradiated by an Enlitech AAA sun simulator, with the intensity calibrated by a Si reference cell. The active area of the solar cell was 0.06 cm$^2$. The forward photocurrent–voltage scans were measured from –0.2 V to 1.2 V and the reverse scans were from 1.2 V to –0.2 V, both at a scan rate of 40 mV s$^{-1}$. We conducted reverse scan before the forward scan. For the electrochemical impedance spectroscopy measurement, a STAT-I-400 potentiostat of Metrohm LTD was used. The signal was measured starting from 1 MHz down to 1 Hz over 50 frequency points, using a potential amplitude of 20 mV, under white light (100 mW cm$^{-2}$) at open-circuit voltage condition.

## CW-PPPc measurement

Two continuous waveform and constant-powered laser diodes were used as the pump and push sources in the CW-PPPc setup. The pump and push beam were combined together via a dichroic mirror. Then the collinear pump and push beam was focused on the device with an overlapping spot size of 0.0003 cm$^2$. For $\triangle J_{IR}$ measurement, an optical chopper was placed in the push path (before combination with the pump path) and its frequency was fixed at 717 Hz by a chopper controller (MC2000B, Thorlabs). For $J_{pump}$ measurement, a similar optical chopper was placed at the pump path (before combination with the push path). The modulated current from device at short circuit was then recorded by a lock-in amplifier (SR830, Stanford Research Systems).

The intensity-dependent measurement was conducted with an 808-nm pump beam (CPS808, Thorlabs) and a 980-nm push beam (CPS980, Thorlabs) at room temperature. The intensity of pump and push beam was controlled by a motorised filter wheel (FW212C, Thorlabs). The experiment was conducted under ambient atmosphere.

The thermal-dependent measurement was conducted with a 450-nm pump beam (220 mW cm$^{-2}$, CPS450, Thorlabs) and a 1550-nm push beam (5300 mW cm$^{-2}$, LDM1550, Thorlabs). This could minimise the impact of absorption coefficient change of pump beam and ensure the push beam could not directly excite carriers from VB, under low temperatures. The device was placed at a cryostat (HFS600E-PB4, Linkam) [equipped with a liquid nitrogen cooling module (LNP96, Linkam)] to control its temperature from 240 K to room temperature.

Nitrogen was fully filled in the chamber to prevent any potential degradation of perovskites. This system can also control the temperature above room temperature, which was used to heat up the device from room temperature to 80 °C (and back to room temperature) for the temperature-dependent photocurrent measurement (Fig. 2c).

## ns-PPPc measurement

Two pulse lasers were used as the pump and push sources in the ns-PPPc setup. The pump pulse (800 nm, 40 fs, 4000 Hz) was provided by a Ti:sapphire regenerative amplifier (Astrella, Coherent). The push pulse (1064 nm, 10 ns, 4000 Hz) was provided by a Picosecond Nd:YVO4 laser system (piccolo AOT1, Innolas Laser). The pump and push pulses were combined together via a dichroic mirror. Then the collinear pump and push pulses were focused on the device with an overlapping spot size of 0.003 cm$^2$. We equipped a motorised filter wheels (FW212C, Thorlabs) to vary the fluence of pump or push pulses. The time delay between the pump and push pulses was achieved by an electrical delay generator (DG645, Stanford Research Systems). For $\triangle J_{IR}$ measurement, an optical chopper was placed at the push path (before combining with the pump path) and its frequency was fixed at 717 Hz by a chopper controller (MC2000B, Thorlabs). For $J_{pump}$ measurement, such an optical chopper was placed at the pump path (before combining with the push path). The modulated device current was then recorded by a lock-in amplifier (MFLI 500 kHz, Zurich Instruments). The experiment was conducted under ambient atmosphere.

## ns-TAS measurement

Two pulse lasers were used as the pump and probe sources in the ns-TAS setup. The pump pulse (532 nm, 10 ns, 4000 Hz) was generated through second harmonic generation from an ND:YVO4 laser (piccolo AOT1, Innolas Laser). This pump pulse was coupled to a fibre and recollimated at the output side, focusing to the sample with a 50 mm plan-convex lens and having a spot size of 0.008 cm$^2$. The probe pulse (peak at 800 nm, 40 fs, 4000 Hz) was provided by the Ti:sapphire Laser (Coherent Astrella). The probing wavelength of 755–845 nm was achieved by significantly reducing its intensity, which allowed us to access the ground state bleach signal of perovskites. The probe pulse was focused on the sample through a 150-mm spherical mirror and then recollimated and coupled to a fibre as input of the spectrometer. The time delay between the pump and probe pulse was achieved by an electrical delay generator (DG645, Stanford Research Systems). To detect the probe spectrum, a self-made spectrometer was built with N-F2 prism and GaInAs CCD cameras (Entwicklungsbuero EB Stresing). An optical chopper (2000 Hz) was synchronised with the camera trigger and placed on the pump path. The experiment was conducted under ambient atmosphere.

## Reporting summary

Further information on research design is available in the Nature Portfolio Reporting Summary linked to this article.

# Data availability

The data that support the findings of this study are provided in the main text and the Supplementary Information. The original data are available from the corresponding author upon request.

# Code availability

The Driftfusion code and parameter setting used for the drift-diffusion model is available in: https://github.com/barnesgroupICL/Driftfusion[63]. The modified code for the drift-diffusion model and the code for the kinetic model are available from the corresponding author upon request.

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

## Acknowledgements

The authors thank Prof. James R. Durrant and his lab at Imperial College London for providing facilities for sample fabrication. We thank Lucy Hart for discussion on the trapping, recombination and deptrapping kinetics. Z.C. is a Marie Skłodowska-Curie Postdoctoral Fellow (Project No.: 101064229) funded by UK Research and Innovation (Grant Ref.: EP/X027465/1). A.A.B. acknowledges support from the Royal Society and Leverhulme Trust. This work was supported by the European Research Council under the European Union's Horizon 2020 research and innovation programme (Grant Agreement No. 639750).

## Author contributions

J.P., B.H., Z.C., and Z.M. conducted the CW- and ns-PPPc experiments. Z.C. and Z.S. established the CW-PPPc kinetic model. H.N. conducted the ns-TAS experiment. H.N. and P.R.F.B. establish the drift-diffusion model. T.Z., W.X., and G.M. fabricated PeSCs for this project. T.Z. carried out the perovskite film and device characterisations. M.C. carried out the PL quantum yield measurement. D.N. and N.G. carried out the electrochemical impedance spectroscopy measurement and analysis. X.L. carried out the XPS measurement and analysis. Z.C., J.P., B.H., and A.A.B. analysed the data and wrote the manuscript. A.A.B., F.G., P.R.F.B., and S.A.H. led the project. All authors contributed to the manuscript.

## Competing interests

The authors declare no competing interests.
