## [Peer Review File · Nature Communications]

Operando Dynamics of Trapped Carriers in Perovskite Solar Cells Observed via Infrared Optical Activation SpectroscopyREVIEWER COMMENTS

Reviewer #1 (Remarks to the Author):

This report investigate the trapping carrier dynamics in-situ the perovskite solar cells, which is one of the most critical questions in perovskite photovoltaic community. The authors use the pump-IR probe either steady-state or transient fashions, nanosecond transient absorption dependence on laser intensity, frequency to unravel the dynamics. I truly believe that the authors have very good motivation, however, in term of the novelty of characterization techniques, and detail parameter study, I would suggest the authors have in-depth insight, although there are some theoretical simulation features.

1. For the novelty of techniques, pls refer to the review "Ultrafast Spectroscopy with Photocurrent Detection: Watching Excitonic Optoelectronic Systems at Work" J. Phys. Chem. Lett. 2016, 7, 2, 250–258.

2. As the main idea is about the in-situ device characterization, the electrical field and temperature dependence would be the most critical parameter to have more insight of carrier drift-diffusion dynamics. (Although the this report includes temperature in the CW pump-probe, the important and desirable time-resolved pump-probe dependence are missing)

As I deeply appreciate this work which focus on the trapping carriers, but the weight of this work that carry not reach the level to be published in this journal.

Reviewer #2 (Remarks to the Author):

This manuscript by Pan et al. studies the photocarrier trapping dynamics in perovskite solar cells, which is a critical and timely topic. They utilized a novel optical pump-IR push-photocurrent spectroscopy to probe the concentration of trapped photocarriers in perovskite solar cells with or without a surface passivation layer. They claim that their technique can provide information on trap density and trapped carrier lifetime and they see a clear difference between pristine and surface-passivated samples. My main technical concern is that whether the IR push technique is really sensitive to the trapped charge carriers. Given the large mismatch between the IR push photon energy (> 1 eV) and the defect activation energy (280 meV estimated in this paper), the IR push pulse can well excite free photocarriers (generated by the pump) to higher energy levels which can also contribute to the measured photocurrent. If the authors cannot uniquely pinpoint the measured signal to the charge traps, all their

subsequent discussions are meaningless. In addition to this major concern, I have the following technical comments:

1. I am not convinced that the chopper-frequency-dependent CW measurement can provide accurate estimation of the trapped carrier lifetime. There are many other processes that can contribute to the frequency dependence of the photocurrent signal, for example any capacitive response in the device during the photocarrier transport process. In fact, given the low chopper frequency and the unphysically long trapped carrier lifetime (is there any justification of the ms level lifetime, e.g. from literature?), I believe capacitances associated with the photocarrier transport are more likely causes of the observed frequency response.

2. The authors should also show the phase signal from their lock-in measurements. An increase in the amplitude signal from the lock-in amplifier does not necessarily mean an increase in the actual photocurrent, unless a positive phase is also measured. This small detail is important here since the authors ruled out the IR-induced thermal effect based on the fact the increased temperature leads to a reduced photocurrent. If the lock-in phase is not carefully checked, however, this claim is not valid.

3. The heating effect of both the pump and the IR beam should be more carefully examined. Either temperature measurement or thermal simulation should be provided to make sure beam-induced temperature variation does not interfere with data interpretation.

4. What is the thickness of the active device layer and how does that compare to the optical penetration depth of the pump and the IR beam? This is critical since the illumination is from the side without surface passivation. The optical penetration depth should be evaluated to see whether the IR push beam can directly influence the passivated surface.

5. Error bars should be provided wherever possible in the presented experimental data.

Reviewer #3 (Remarks to the Author):

The authors present work about operando studies of perovskite solar cells with a novel optical spectroscopy method. The target is the determination of the surface versus bulk charge carrier trapping dynamics. While the dynamics parts has some new insights, their general value is rather limited due to the present shortcomings of the manuscript. In detail, the problems are:

1. The work misses a link of the observed trap states with the structure of the active perovskite layer. For a high impact journal such correlation is mandatory and needs to be added to enable publication.
2. Operando studies are highly demanding but only useful in case the studied devices have a relevant power conversion efficiency. Looking to the device data shown in table S1, one can question this relevance, which leads to the direct question, why the authors did not study higher efficiency devices? In particular, for a first work and high profile journal, it is mandatory to have efficiencies close to the state-of-the-art. Thus, the presented work is better suited for a more specialized journal, which has a focus on method development and not on the general findings for the perovskite community.
3. What is missing, is a structure determination to make the reader understand what is the type of perovskite active layer established in the present work? Obviously, there is shortcomings but unclear to some extent, which is the structural origin and then causing the observed trap states.
4. Unfortunately, the experimental condition of the solar cell operation are not clear. Was the operation under inert atmosphere or under vacuum? From the used set-up one might suppose vacuum condition. As seen in the literature (Degradation mechanisms of perovskite solar cells under vacuum and one atmosphere of nitrogen, *Nature Energy* 6, 977-986, 2021), such question is highly relevant and needs to be well discussed.
5. Charge-carrier trapping and radiative recombination in metal halide perovskite semiconductors as discussed in *Adv. Funct. Mater.* 30, 2004312 (2020) needs to be taken into better account in the present work.

As a consequence, the present version of the manuscript is not at all suited for publication in any high profile journal and should be moved to a more specialized journal dealing with technical developments and methods.

Point-by-point response to reviewers' comments

Reviewer #1:

This report investigates the trapping carrier dynamics in-situ the perovskite solar cells, which is one of the most critical questions in perovskite photovoltaic community. The authors use the pump-IR probe either steady-state or transient fashions, nanosecond transient absorption dependence on laser intensity, frequency to unravel the dynamics. I truly believe that the authors have very good motivation, however, in term of the novelty of characterization techniques, and detail parameter study, I would suggest the authors have in-depth insight, although there are some theoretical simulation features.

Reply:

We thank the Reviewer for their kind evaluation of our work. We followed their suggestions to further improve the scientific quality of our manuscript.

1. For the novelty of techniques, pls refer to the review "Ultrafast Spectroscopy with Photocurrent Detection: Watching Excitonic Optoelectronic Systems at Work" *J. Phys. Chem. Lett.* 2016, 7, 2, 250–258.

Reply:

We thank the Reviewer for their suggestion. We cited the suggested paper as reference 36 and added the following discussion in our revised manuscript:

On page 4:

“By taking advantage of photocurrent detection, PPPc can successfully overcome the limitations of conventional optical methods.³⁶ These methodologies also have the advantage of allowing trapped carriers to be monitored selectively and sensitively with relatively low optical-excitation intensities or fluences.”

In the Reference section,

[36] Bakulin, A. A., Silva, C. & Vella, E. Ultrafast spectroscopy with photocurrent detection: watching excitonic optoelectronic systems at work. *J. Phys. Chem. Lett.* 7, 250–258 (2016).

2. As the main idea is about the in-situ device characterization, the electrical field and temperature dependence would be the most critical parameter to have more insight of carrier drift-diffusion dynamics. (Although the report includes temperature in the CW pump-probe, the important and desirable time-resolved pump-probe dependence are missing)

Reply:

We thank the Reviewer for their constructive suggestions.

We also realise that the temperature-dependent experiment on time-resolved PPPc can provide us with a more in-depth understanding of the dynamics of band-edge carriers and trapped carriers. We therefore followed the Reviewer's recommendation and conducted such experiments in both the pristine and surface-passivated device from 300 K to 240 K. Below 230 K, there is a phase change of FAPbI₃, which would lead to a change of nature of traps (Adv. Funct. Mater. 2020, 30, 2004312). In the current research, we would not like to involve such a complicated situation. Please also note that the devices we measured were an optimised version with an ultra-thin PMMA layer on top of the SnO₂ layer, following the suggestion by Reviewer 3 that we should study better devices with efficiency closer to the state-of-the-art (detailed can be found in the response to Reviewer 3, point 2).

The new results are shown in Figures S16 of the revised manuscript (also shown below). The main features of the dynamics stay the same and can be described by the model presented in the original manuscript. We observed a short-delay time spike in the devices with PMMA, which we attribute to the instant optical effect between pump and push pulses, rather than actual trapped carrier dynamics. The effect is more obvious when the PPPc signal is relatively small and the device structure is more complex.

We also observed slower rises in both pristine and surface-passivated devices when decreasing the temperature below 260 K, indicating that the reduction of carrier diffusion has a substantial contribution to charge transport at these conditions (Mater. Horiz. 2020, 7, 397–410). An alternative explanation can be the higher positive charge density built up by trapped holes at the perovskite/Spiro-OMeTAD interface due to the reduction of trap-assisted recombination rates and more efficient trapping at lower temperatures. However, the recombination processes substantially slow down, elevating the baseline and making amplitude analysis problematic when high repetition rates are used.

To reflect this, we have added the following discussion and figures in our revised manuscript:

On page 14,

“Using optimised devices as a model system, we conducted temperature-dependent ns-PPPc measurement in 300–240 K region (Figure S16). The main features of the dynamics at room temperature shown in Figure S16a and S16b stay the same and can be described by the model

presented above. The recombination processes substantially slow down, elevating the baseline and making amplitude analysis problematic when high repetition rates are used. We also observed slower rise in both pristine and surface-passivated devices when decreasing the temperature below 260 K, indicating that the reduction of carrier diffusion has a substantial contribution to charge transport at these conditions.⁴ An alternative explanation can be the higher positive charge density built up by trapped holes at the perovskite/Spiro-OMeTAD interface due to the reduction of trap-assisted recombination rates and more efficient trapping at lower temperatures.”

In Supporting Information:

Figure S16. Temperature-dependent results based on different devices. Trapped carrier dynamics of optimised (a) pristine device and (b) surface-passivated device, under pump fluence of $3.3 \mu\text{J cm}^{-2}$ and push fluence of $170.8 \mu\text{J cm}^{-2}$. Note that the spikes shown in both cases are due to the instant optical effect between pump and push pulses, rather than the occurrence of real trapped carrier dynamics.

In addition, we fully agree with the reviewer that investigating the electric-field-dependent behaviour of trapped carriers is also crucial to providing more in-depth understanding. However, we note that since perovskite solar cells are well established to be rich in mobile ionic defects, electric fields in the bulk of the perovskite are screened by redistribution of these defects so even if there was field dependence, it would not be observed. Fields are present at the interfaces however, and since

we have shown much of the trapped charge accumulates at the interfaces, this is indeed an interesting question.

Also, the electric-field-induced ion migration (Acc. Chem. Res. 2016, 49, 286–293) can potentially change the distribution, types and even number of traps. This would require careful consideration and control to ensure that the results are not skewed by such effects. Or, we should also consider ion migration (and its impact) together when analysing trapped carrier dynamics.

Moreover, PPPc is based on current detection. From our experience, applying voltages to the device significantly increases the basic noise for current detection and can bury the desired PPPc signal. This would require extra strategies or circuit designs to eliminate the electrical noise generated by the voltage sources.

Therefore, conducting electric-field-dependent experiments requires more complex experimental setups and result interpretation, which would require a new project dedicated to this investigation. In such a case, we decided to focus on the temperature-dependent behaviour of carriers and kept the field-dependence investigation for future studies.

As I deeply appreciate this work which focus on the trapping carriers, but the weight of this work that carry not reach the level to be published in this journal.

Reply:

We thank the reviewer for their honest opinion. We hope that the additional results provided make the revised manuscript suitable for publication in Nature Communication.

Reviewer #2:

This manuscript by Pan et al. studies the photocarrier trapping dynamics in perovskite solar cells, which is a critical and timely topic. They utilized a novel optical pump-IR push-photocurrent spectroscopy to probe the concentration of trapped photocarriers in perovskite solar cells with or without a surface passivation layer. They claim that their technique can provide information on trap density and trapped carrier lifetime and they see a clear difference between pristine and surface-passivated samples.

Reply:

We thank the Reviewer for their kind evaluation of our work.

My main technical concern is that whether the IR push technique is really sensitive to the trapped charge carriers. Given the large mismatch between the IR push photon energy (> 1 eV) and the defect activation energy (280 meV estimated in this paper), the IR push pulse can well excite free photocarriers (generated by the pump) to higher energy levels which can also contribute to the measured photocurrent. If the authors cannot uniquely pinpoint the measured signal to the charge traps, all their subsequent discussions are meaningless. In addition to this major concern, I have the following technical comments:

Reply:

The Reviewer raises an important question about the nature of PPPc signal and we agree with the Reviewer that the validity of the PPPc technique for trap state measurement should be discussed more clearly in the manuscript. PPPc signal corresponds to the difference in total number of charges collected with and without assistance of push pulse. For this reason, PPPc is not sensitive to the energy (hot/cold) of the carriers or their mobility. Only the (trap-assisted) recombination processes, that affect the total amount of charge extracted, are observed. This makes PPPc selectively sensitive to the population of trap states that are precursors for recombination dynamics.

We add the following explanations to the revised manuscript:

On page 8,

“We note that band-edge free carriers can also be excited by IR push beam to higher energy states, however, this would cause no significant change in photocurrent because i) exciting free carriers to hot states does not change the number available for extraction from the device, and ii) hot carrier trapping is typically not observed in 3D perovskites.^{41,42} Therefore, PPPc is a highly sensitive and selective method for monitoring carriers trapped in states that act as recombination centres in the device (further discussion of the origin of PPPc signal can be found in Supplementary text 2).”

In the Reference section,

*[41] Carwithen, B. P. et al. Confinement and exciton binding energy effects on hot carrier cooling in lead halide perovskite nanomaterials. *ACS Nano* **17**, 6638–6648 (2023).*

*[42] Hopper, T. R. et al. Ultrafast intraband spectroscopy of hot-carrier cooling in lead-halide perovskites. *ACS Energy Lett.* **3**, 2199–2205 (2018).*

“Supplementary text 2: Origin of PPPc signal

Figure S17 illustrates the fundamental operating principle of PPPc. Under optical illumination a photocurrent density $J_{without-IR}$ is extracted from the device. This photocurrent is composed from thermalised (cold) free carriers so that $J_{without-IR} = J_{cold-carrier}$. When the IR push beam illuminates the device and a new photocurrent density $J_{with-IR}$ is observed. Some trapped carriers are optically detrapped by the IR and contribute an additional component $J_{bound-carrier}$ to the photocurrent density. Band-edge free carriers will also be excited to higher energy states and some these will also contribute an amount $J_{hot-carrier}$ to $J_{with-IR}$. However, this reduces the population of thermalised carriers contributing to $J_{with-IR}$ by a similar amount so that $J_{cold-carrier} = J_{without-IR} - J_{hot-carrier}$. Since most hot carriers quickly cool back to band-edge to become cold carriers typically within 1 ps,³ much faster than the time scale for charge collection at the contacts (~100 ns, see Figure 4), so we would expect $J_{hot-carrier}$ to be very small. Our previous research has also shown that hot carrier trapping generally is not present in 3D perovskites, so we would not expect its occurrence here.^{4,5} Therefore, the additional photocurrent density induced by the IR beam is given by:

$$\begin{aligned}\Delta J_{IR} &= J_{with-IR} - J_{without-IR} \\ &= [J_{hot-carrier} + (J_{without-IR} - J_{hot-carrier}) + J_{bound-carrier}] - J_{without-IR} \\ &= J_{bound-carrier}\end{aligned}\tag{S2.1}$$

The high selectivity of PPPc, which is only sensitive to bound species in the device and make unbound species undetectable, has made it a valuable tool in the investigation of organic solar cells over the past decade.⁶ In organic solar cells, bound species typically refer to excitons and traps, while in 3D perovskite solar cells, the bound species are solely traps since negligible excitons are present due to their small exciton binding energy.”

Figure S17. Schematic diagram showing the hole extraction (for current generation) without and with IR push. All the arrows indicate physical processes in terms of holes.

[3] Villamil Franco, C. *et al.* Exciton cooling in 2D perovskite nanoplatelets: rationalized carrier-induced stark and phonon bottleneck effects. *J. Phys. Chem. Lett.* **13**, 393–399 (2022).

[4] Hopper, T. R. *et al.* Hot carrier dynamics in perovskite nanocrystal solids: role of the cold carriers, nanoconfinement, and the surface. *Nano Lett.* **20**, 2271–2278 (2020).

[5] Hopper, T. R. *et al.* Ultrafast intraband spectroscopy of hot-carrier cooling in lead-halide perovskites. *ACS Energy Lett.* **3**, 2199–2205 (2018).

[6] Zhang, J. *et al.* Efficient non-fullerene organic solar cells employing sequentially deposited donor–acceptor layers. *J. Mater. Chem. A* **6**, 18225–18233 (2018).”

1. I am not convinced that the chopper-frequency-dependent CW measurement can provide accurate estimation of the trapped carrier lifetime. There are many other processes that can contribute to the frequency dependence of the photocurrent signal, for example any capacitive response in the device during the photocarrier transport process. In fact, given the low chopper frequency and the unphysically long trapped carrier lifetime (is there any justification of the ms level lifetime, e.g. from literature?), I believe capacitances associated with the photocarrier transport are more likely causes of the observed frequency response.

Reply:

We agree with the reviewer that when using various chopper frequencies to access the time-related response, the resistor-capacitor (RC) response may be a limiting factor for the observed timescale.

To show that in our case RC time is not the limiting factor we have (i) performed the modulation frequency dependent measurement over a broader range (to 4000 Hz) and (ii) directly measured RC constant using electrochemical impedance spectroscopy.

First, we study the frequency-dependent J_{Pump} , which allows us to access the characteristic time constant of band-edge free carriers. Figure R1 shows experimental data and Cole-Cole fitted results. The fitted characteristic time constant of free carriers in pristine and surface-passivated devices are $21.8 \mu\text{s}$ and $19.7 \mu\text{s}$, respectively, similar to those lifetimes measured by transient photocurrent (Nat. Commun. 2021, 12, 466). Considering this is a convoluted result, it suggests that the time of RC respond is faster than the values acquired from the fits.

Figure R1. ΔJ_{Pump} signal as a function of modulation frequency. The solid lines indicate the fitted results according to Equation 2 in the main text.

To cross-check, we conduct electrochemical impedance spectroscopy studies on both pristine and surface-passivated devices, as shown in Figure S9 (also shown below). The extracted resistance and capacitance are 15.2Ω and 19 nF for pristine and 24Ω and 9.6 nF for surface-passivated devices, corresponding to RC time constant of $0.29 \mu\text{s}$ and $0.23 \mu\text{s}$ for pristine and surface-passivated PeSCs, respectively. Such values are significantly smaller than the extracted characteristic time constant of 51.5 ms and 33.5 ms for pristine and surface-passivated devices (Figure 3c), respectively. Hence, we consider the impact of RC response to be negligible for the characteristic time constant estimation in our case. However, although the characteristic time constant, which relates to trapped carriers, agrees with the ms-scale recombination lifetime that had been reported by transient photovoltage experiments previously (ACS Energy Lett. 2019, 4, 1784–179; J. Mater. Chem. A, 2019, 7, 13777–13786), we agree with the reviewer that we should better discuss the understanding of this extracted time constant.

We therefore make a major revision to the discussion and added a new figure of Figure S9 in our

revised manuscript:

On pages 9–10,

“The dependence of $\Delta J_{IR}/J_{Pump}$ on IR chopper modulation frequency is shown in Figure 3c. It is apparent that the signal decreases as the frequency increases for both samples, while the phase shift of the signal (Figure S3) remained constant and close to zero over the entire frequency range (37 – 4000 Hz). A frequency-dependent phase shift would be expected if the variation in the CW-PPPC signal were induced by IR heating, which we did not observe. To determine a characteristic timescale of the process resulting in the frequency-dependent signal, we fitted the data with the Cole-Cole equation commonly used to analyse the chopper frequency-dependent measurements:^{43,44}

$$\frac{\Delta J_{IR}}{J_{Pump}} = \frac{\left(\frac{\Delta J_{IR}}{J_{Pump}}\right)_0}{1 + (i2\pi f\tau)^\alpha} \quad (2)$$

where the $\left(\frac{\Delta J_{IR}}{J_{Pump}}\right)_0$ is the projected steady-state $\Delta J_{IR}/J_{Pump}$ value at 0 Hz; f is the chopper modulation frequency; τ represents the characteristic time constant for the process and the exponent α reflects the dispersity or range of timescales of the underlying process (if $\alpha = 1$ the process is non-dispersive and the process is increasingly dispersive as $\alpha \rightarrow 0$). For the pristine device $\tau = 52$ ms and $\alpha = 0.52$; for the surface passivated device a slightly faster but considerably more dispersive process was observed with $\tau = 34$ ms and $\alpha = 0.28$. These timescales are considerably longer than the series resistor-capacitor (RC) time constant of the devices which put a lower limit on the time for charge extraction (determined from electrochemical impedance spectroscopy to be $RC = 0.29 \mu\text{s}$ and $0.23 \mu\text{s}$ for pristine and surface-passivated PeSCs, respectively, see Figure S9). We tentatively suggest these highly dispersive time constants reflect the influence of recombination via a subset of trap states with a wide range of depths or capture cross-sections in the device, particularly those at the interfaces. At the interfaces where the majority free carrier species is at a minimum due to repulsion by accumulated mobile ionic defects, recombination times can be very long. Some of the dispersion may also result from a wide range of IR absorption cross-section for the trapped carriers, some of which may be so weakly absorbing that they will only contribute to $\Delta J_{IR}/J_{Pump}$ when illuminated for a sufficiently long time. The increase in dispersity observed in the passivated device appears to indicate a broader and less modal distribution of trap depths, capture cross-sections (with associated recombination times), or absorption coefficients, remaining after surface passivation. This can be attributed to the less domination of surface traps in the total trap states after the passivation process.”

In the Methods section,

“For the electrochemical impedance spectroscopy measurement, a STAT-I-400 potentiostat of

Metrohm LTD was used. The signal was measured starting from 1 MHz down to 1 Hz over 50 frequency points, using a potential amplitude of 20 mV, under white light (100 mW cm^{-2}) at open-circuit voltage condition.”

In Supporting information,

Figure S9. RC constant extraction from pristine and surface-passivated PeSCs measured by electrochemical impedance spectroscopy under white light (100 mW cm^{-2}) at open-circuit voltage condition. (a) Nyquist plots and (b) capacitance as a function of frequency. The series resistance (R_s) of the devices was extracted from the Nyquist plots **a while the geometric capacitance (C_g) was extracted at 1 MHz from **b**. The calculated RC time constants are $0.29 \mu\text{s}$ and $0.23 \mu\text{s}$ for pristine and surface-passivated PeSCs, respectively.**

2. The authors should also show the phase signal from their lock-in measurements. An increase in the amplitude signal from the lock-in amplifier does not necessarily mean an increase in the actual photocurrent, unless a positive phase is also measured. This small detail is important here since the authors ruled out the IR-induced thermal effect based on the fact the increased temperature leads to a reduced photocurrent. If the lock-in phase is not carefully checked, however, this claim is not valid.

Reply:

We agree with the reviewer that it is necessary to show the X, Y, and R values of lock-in to clearly confirm the origin of IR-induced photocurrent. We indeed measured all components of the signal and we now revise Figure 2b and add Figure S11 in the revised manuscript (also shown below) to report the X, Y, and R values for both CW- and ns-PPPC. According to these figures, illumination by “Push Only” or “Pump+Push” does not change the phase of the signal. Y signal stays close to zero, while the R signal mainly follows the X signal at the modulation frequency of 717 Hz used. Moreover, we measure the intensity-dependent X, Y, and R values (Figure S3, also shown below), which further

confirms the above conclusion, and no obvious phase change is observed.

In addition, the observed 90-degree phase change in ns-PPPC measurements after 1×10^5 ns is expected and attributed to the comparable time scale of the electrical delay and the modulation period of the chopper. This effect is accounted for and does not affect the results, when the total amplitude R value is presented.

Figure 2. (b) ΔJ_{IR} response of $FA_{0.99}Cs_{0.01}PbI_3$ device under different illumination conditions in CW-PPPC measurement. The ΔJ_{IR} only responded when both pump and push beams illuminated the device simultaneously.

Figure S3 Lock-in amplifier signal as a function of modulation frequency in CW-PPPC measurement. X, Y, and R values of (a) pristine and (b) surface-passivated devices, as well as corresponding phases of (c) pristine and (d) surface-passivated devices. The phases remains approximately constant across all frequencies.

Figure S11. ΔJ_{IR} response of $\text{FA}_{0.99}\text{Cs}_{0.01}\text{PbI}_3$ device under 800-nm pump ($5.1 \mu\text{J cm}^{-2}$) and 1064-nm push ($170.8 \mu\text{J cm}^{-2}$) in ns-PPPC measurement. The observed 90-degree phase change in time-resolved PPPc measurement after 1×10^5 ns is expected and attributed to the comparable time scale of the electrical delay and the modulation period of the chopper. This effect is accounted for and does not affect the results, when the total amplitude R value is presented.

Accordingly, we have added the following discussion in our revised manuscript:

On page 7,

“By comparing X (in-phase) and Y (out-of-phase) signal from the lock-in amplifier in Figures 2b and S3, we conclude that there is no significant change of phase at various chopper modulation frequencies.”

On page 11,

“In agreement with CW-PPPC observations, the devices appeared to be in a first-order recombination regime (Figure S10), and the R value (i.e., total IR-induced current) of the detected signal provides a good description of population dynamics (Figure S11).”

3. The heating effect of both the pump and the IR beam should be more carefully examined. Either temperature measurement or thermal simulation should be provided to make sure beam-induced temperature variation does not interfere with data interpretation.

Reply:

We agree that both pump and IR beam can change the sample temperature and we therefore performed a quantitative evaluation of the beam-induced heating effects.

The heat generated by the pump beam do not contribute to the ΔJ_{IR} because the chopper was placed in the push path, and the lock-in amplifier only recorded changes induced by the IR beam. Therefore, the heat generated by the pump beam just affect the equilibrium temperature of the sample, probably raising it slightly above the room temperature.

To quantify how the IR push affects the temperature of perovskite in CW experiments, we estimate its temperature increase (ΔT) resulting from the absorption of IR beam by the following equation:

$$\Delta T = \frac{Q}{m \cdot c} = \frac{A \cdot P_{IR} \cdot \frac{1}{2f_{IR}}}{\rho \cdot V \cdot c} = \frac{A \cdot P_{IR}}{\rho \cdot d \cdot S \cdot c \cdot 2f_{IR}}$$

Here:

Q is the absorbed energy of IR beam when the chopper is on;

m is the mass of the perovskite;

c is the specific heat capacity ($308 \text{ J kg}^{-1} \text{ K}^{-1}$, *Nat. Commun.* 2021, 12, 3348);

A is the percentage of the light absorbed by photogenerated carriers in the active layer of the device (measured using TA spectroscopy to be below 0.15%);

P_{IR} is the maximum power of 980-nm IR beam in our research (i.e., 1.3 mW);

ρ is the density of perovskite ($4.1 \times 10^{-3} \text{ kg cm}^{-3}$, *Inorg. Chem.* 2013, 52, 9019–9038);

V is the volume of perovskite under IR illumination, which is determined by the product of d (the distance of light passing through the perovskite layer, 1600 nm) and S (beam size of IR light, 0.0012 cm^2);

f_{IR} is the chopper frequency (717 Hz).

In this case, even assuming no heat flows out of the active layer during the illumination, the maximum IR power used increases the temperature of perovskite by 0.0056 K. According to Figure 2c, such a temperature increase could cause a decrease in photocurrent density by $5.5 \times 10^{-6} \text{ mA cm}^{-2}$, which is over 2 orders smaller than the typical PPPc signal ($\sim 1 \times 10^{-3} \text{ mA cm}^{-2}$) at the corresponding pump and push intensity (Figure S4). Note that this estimation does not consider contributions from substrate heating by the IR beam, because the transfer of heat from the substrate to the perovskite would have a time lag which would manifest itself as a decreased out-of-phase (Y) component of the signal (according to Figure 2c) at lower frequencies which, however, was not observed in Figure S3. Hence, we consider that heat generated by the IR push has a negligible effect on the PPPc results in our study.

Therefore, we add the following discussion in our revised manuscript:

On page 7,

“Moreover, we estimate that any change in photocurrent induced by IR-heating is at least two orders smaller than the PPPc signal under the same illumination conditions (Supplementary Text 1), further indicating that IR heating has a negligible impact on our results.”

In Supporting Information,

“Supplementary text 1: Estimation of the impact of IR heat on photocurrent difference.

To estimate how the IR push could affect the temperature of perovskite in CW experiments, we estimate its temperature increase (ΔT) resulting from the absorption of IR beam by the following equation:

$$\Delta T = \frac{Q}{m \cdot c} = \frac{A \cdot P_{IR} \cdot \frac{1}{2f_{IR}}}{\rho \cdot V \cdot c} = \frac{A \cdot P_{IR}}{\rho \cdot d \cdot S \cdot c \cdot 2f_{IR}} \quad (S1.1)$$

Here:

Q is the absorbed energy of IR beam when the chopper is on;

m is the mass of the perovskite;

c is the specific heat capacity of perovskite ($308 \text{ J kg}^{-1} \text{ K}^{-1}$);¹

A is the percentage of the light absorbed by photogenerated carriers in the active layer of the device (measured using TAS to be below 0.15%);

P_{IR} is the maximum power of 980-nm IR beam in our research (i.e., 1.3 mW);

ρ is the density of perovskite ($4.1 \times 10^{-3} \text{ kg cm}^{-3}$);²

V is the volume of perovskite under IR illumination, which is determined by the product of d (the distance of light passing through the perovskite layer, 1600 nm) and S (beam size of IR light, 0.0012 cm^2);

f_{IR} is the chopper frequency (717 Hz).

In this case, even assuming no heat flows out of the active layer during the illumination, the maximum IR power used increases the temperature of perovskite by 0.0056 K. According to Figure 2c, such a temperature increase could cause a decrease in photocurrent density by $5.5 \times 10^{-6} \text{ mA cm}^{-2}$, which is over 2 orders smaller than the typical PPPc signal ($\sim 1 \times 10^{-3} \text{ mA cm}^{-2}$) at the corresponding pump and

push intensity (Figure S4). Note that this estimation does not consider contributions from substrate heating by the IR beam, because the transfer of heat from the substrate to the perovskite would have a time lag which would manifest itself as a decreased out-of-phase (Y) component of the signal (according to Figure 2c) at lower frequencies which, however, was not observed in Figure S3. Hence, we consider that heat generated by the IR push has a negligible effect on the PPPc results in our study.’

4. What is the thickness of the active device layer and how does that compare to the optical penetration depth of the pump and the IR beam? This is critical since the illumination is from the side without surface passivation. The optical penetration depth should be evaluated to see whether the IR push beam can directly influence the passivated surface.

Reply:

Following Reviewer’s suggestion, we have highlighted the actual value of the perovskite thickness in both the revised Figure 1a and Methods section of the manuscript, as shown below.

Figure 1. Device properties of FA_{0.99}Cs_{0.01}PbI₃ PeSCs. (a) Device architecture of the surface-passivated device. The pristine device has a similar architecture but without the OAI layer. (b) Current density versus voltage of the pristine and surface-passivated devices measured during forward and reverse voltage sweeps at a scan rate of 40 mV s⁻¹. (c) Configuration of the surfaces of pristine and surface-passivated perovskites. The OA chains mount to the perovskite surface via their NH₃⁺ groups, which also passivate the FA⁺ vacancies simultaneously, while the I⁻ vacancies can be passivated by the I⁻ in OAI.³⁹

In Methods section,

“The 800-nm perovskite layer was then deposited via spin coating the perovskite precursor at 5000

rpm for 30 s.”

We also appreciate the reviewer's comment on the penetration depth of the pump/IR push beams and perovskite thickness. In response, we have checked the intrinsic absorption coefficients of the 800/808-nm pump and 980-nm push beams for FAPbI₃ from literature, which are reported to be $\sim 3 \times 10^3 \text{ cm}^{-1}$ and $\ll 1 \text{ m}^{-1}$, respectively (Nat. Mater. 2020, 19, 1201–1206; J. Phys.: Condens. Matter 2017, 29, 245702). Based on these values, we can infer that the pump has a penetration depth of $\sim 3.33 \text{ }\mu\text{m}$, while the IR push has a much larger penetration depth of $\gg 1 \text{ m}$ (in the absence of free carriers). Therefore, we can conclude that both the pump and IR push beams can reach the top surface of the perovskite layer.

We also note that, although the penetration depth of both beams is much thicker than the thickness of the perovskite layer, we still apply the Beer-Lambert law when calculating the n_{TC} , for more accurate results, as detailed in Supplementary Text 3.

We have added the following discussion in Supplementary text 3:

“Note that the intrinsic absorption coefficients of the 800/808-nm pump and 980-nm push beams for FAPbI₃, which are reported from literatures, are $\sim 3 \times 10^3 \text{ cm}^{-1}$ (α_{Pump}) and $\ll 1 \text{ m}^{-1}$, respectively.^{7,8} Based on these values, we can infer that the pump has a penetration depth of $\sim 3.33 \text{ }\mu\text{m}$, while the IR push has a much larger penetration depth of $\gg 1 \text{ m}$ (in the absence of free carriers), suggesting that both of them can reach the top surface of perovskite.”

[7] Wright, A. D. *et al.* Intrinsic quantum confinement in formamidinium lead triiodide perovskite. *Nat. Mater.* **19**, 1201–1206 (2020).

[8] Xie, Z. *et al.* Refractive index and extinction coefficient of NH₂CH=NH₂PbI₃ perovskite photovoltaic material. *J. Phys.: Condens. Matter* **29**, 245702 (2017).

5. Error bars should be provided wherever possible in the presented experimental data.

Reply:

We thank the Reviewer for their suggestion.

We have added error bars, which represent the standard deviation of the data, in Figure 3, Figure S4, and Figure S5 in the revised manuscript, which are also shown below.

Figure 3. Trapped carrier concentration in pristine and surface-passivated $FA_{0.99}Cs_{0.01}PbI_3$ devices. (a) $\Delta J_{IR}/J_{Pump}$ versus IR push intensity of a surface-passivated $FA_{0.99}Cs_{0.01}PbI_3$ PeSC. The solid lines indicate the linearly fitted results. (b) Concentration of trapped carriers as a function of pump intensity. The solid lines indicate the fitted results according to Equation 1. (c) $\Delta J_{IR}/J_{Pump}$ signal as a function of modulation frequency. The solid lines indicate the fitted results according to Equation 2. (d) Arrhenius plot of the temperature-dependent trapped carrier concentration. The solid lines indicate the fitted results according to Arrhenius equation. The error bars represent the standard deviation of the data.

Figure S4. ΔJ_{IR} versus IR push intensity curves of surface-passivated devices in CW-PPPC

measurement. The solid lines indicate the linearly fitted results. *The error bars represent the standard deviation of the data.*

Figure S5. $\Delta J_{IR}/J_{Pump}$ versus IR push intensity of a pristine $FA_{0.99}Cs_{0.01}PbI_3$ device under various pump intensities. The solid lines indicate the linearly fitted results. *The error bars represent the standard deviation of the data.*

Reviewer #3:

The authors present work about operando studies of perovskite solar cells with a novel optical spectroscopy method. The target is the determination of the surface versus bulk charge carrier trapping dynamics. While the dynamics parts has some new insights, their general value is rather limited due to the present shortcomings of the manuscript.

Reply:

We thank the Reviewer for their kind evaluation of our work. We have further improved the material aspect and the presentation of our manuscript to warrant its publication in Nature Communications.

In detail, the problems are:

1. The work misses a link of the observed trap states with the structure of the active perovskite layer.

For a high impact journal such correlation is mandatory and needs to be added to enable publication.

Reply:

We agree with the reviewer that determining the type of trap states in perovskite surface lattice would be highly complementary to the photophysics results. We therefore conducted X-ray Photoelectron Spectroscopy (XPS) to study the perovskite surface's stoichiometry and evaluate OAI's passivation effect. As shown in the following figures which are added as Figures S1b to S1d in the revised manuscript, the ratio between Pb and I is calculated to be 1:1.98 in the pristine device, which is much larger than the ideal stoichiometry between Pb and I of 1:3, suggesting abundant Γ^- vacancies exist at the perovskite surface. After depositing the OAI layer, the ratio between Pb and I becomes 1:2.45. The extra Γ^- supplied from OAI can effectively compensate the halide vacancies. In addition, in the pristine film, the calculated Pb:N ratio is 1: 0.715, which is also larger than their ideal stoichiometry of 1:2 in $\text{FA}_{0.99}\text{CS}_{0.01}\text{PbI}_3$, suggesting the existence of FA^+ vacancies at the perovskite surface. After depositing OAI, an extra N peak at ~ 402.5 eV appears (originated from the NH_3^+ in OAI) and the calculated Pb:N ratio becomes 1:1.06. This result also suggests the compensation of FA^+ vacancies by the mounting of the OA chains at the perovskite surface. Therefore, we consider OAI has a bi-functional passivation on both the FA^+ and Γ^- vacancies simultaneously in the studied devices.

We added the following discussion in our revised manuscript.

On page 5,

“This can be attributed to the simultaneous passivation of FA^+ and Γ^- interface vacancies by the OAI molecules, as shown in Figure 1c and discussed in Figures S1b–S1d.”

In the Methods section,

“XPS measurement (monochromatised Al $K\alpha$ $h\nu = 1486.6$ eV) was carried out using a Scienta-200 hemispherical analyser with dedicated home designed and built spectrometer. Data were calibrated by referencing to Fermi level and Au $4f_{7/2}$ peak position of the Ar^+ ion sputter-clean gold foil. The experimental condition was set so that the full width at half maximum of the clean Au $4f_{7/2}$ line (at the binding energy of 84.00 eV) was 0.65 eV.”

In Supporting Information,

Figure S1. Passivation effect and impact of OAI. (a) PL quantum yield measurement of pristine (Glass/FA_{0.99}Cs_{0.01}PbI₃) and surface-passivated (Glass/FA_{0.99}Cs_{0.01}PbI₃/OAI) films. Both films were excited by a 635-nm CW laser. The PL quantum yields of pristine and surface-passivated perovskite films were 4.9% and 13.1%, respectively. XPS of (b) Pb, (c) I and (d) N in both pristine and surface-passivated films. In the pristine film, the calculated Pb:I ratio is 1:1.98, which is much larger than the ideal stoichiometry of 1:3 in perovskites, suggesting an abundance of I⁻ vacancies on the perovskite surface. After depositing the OAI layer, the ratio between Pb and I becomes 1:2.45. The extra I⁻ supplied from OAI compensates the halide vacancies. In addition, in the pristine film, the calculated Pb:N ratio is 1: 0.715, which is also larger than their ideal stoichiometry of 1:2 in FA_{0.99}Cs_{0.01}PbI₃, suggesting the existence of FA⁺ vacancies at the perovskite surface. After depositing OAI, an extra N peak at ~402.5 eV appears (originated from the NH₃⁺ in OAI) and the calculated Pb:N ratio becomes 1:1.06. This result also suggests the compensation of FA⁺ vacancies by the mounting of the OA chains at the perovskite surface. Therefore, we consider OAI has a bi-functional passivation on both the FA⁺ and I⁻ vacancies simultaneously in the studied devices. X-ray diffraction patterns of (e) pristine and (f) surface-passivated perovskite films. We consider no 2D perovskites formed as only 3D perovskite peaks are observed.

2. Operando studies are highly demanding but only useful in case the studied devices have a relevant power conversion efficiency. Looking to the device data shown in table S1, one can question this relevance, which leads to the direct question, why the authors did not study higher efficiency devices? In particular, for a first work and high profile journal, it is mandatory to have efficiencies close to the state-of-the-art. Thus, the presented work is better suited for a more specialized journal, which has a focus on method development and not on the general findings for the perovskite community.

Reply:

We agree with the reviewer that addressing a wider range of devices including the state-of-the-art PeSCs would emphasise the opportunities provided by the ns-PPPC technique. We therefore added the results on highly efficient devices with 23.2% PCE (see below).

However, the bulk of the study we believe should be still focused on a simpler and more versatile system addressed in the original manuscript. Firstly, we prioritise device stability over high efficiency. For the first work using PPPc to study PeSCs, a series of experiments in both CW- and ns-PPPC need to be conducted in an ambient atmosphere for the same device for weeks. Therefore, it requires the device to have an excellent ambient stability for months without encapsulation. Studied systems also had an excellent light stability to tolerate the laser for days (for fluence-dependent experiments) and an excellent thermal stability to tolerate temperature circulation between 240 K to 350 K (for temperature-dependent measurement. Secondly, we selected this typical perovskite and device architecture to provide general and representative conclusions for the field. The device architecture follows a basic structure of ITO/SnO₂/Perovskite/Spiro-OMeTAD/Au. This may lead to the modest photovoltaic performance but provide a “clean” device system to eliminate any unexpected effects and artifacts. According to the literature, the devices with similar active layer and electrode configuration also achieve typical efficiencies of ~20% PCE (e.g., Nature Energy 2021, 6, 977-986; Nat. Commun. 2018, 9, 3021; Adv. Energy Mater. 2020, 10, 2001759).

To better explain why this type of device was selected, we added the following discussion in the revised manuscript:

On page 5,

“These device composition and architectures were selected for their outstanding ambient (~months), light, and thermal stability without the need for encapsulation as well as for being a typical model system studied across the perovskite photovoltaic community.”

Same time, following the Reviewer's suggestion, we applied the developed methodology to more efficient devices. For this, we further optimise the devices by adding an ultra-thin PMMA layer above SnO₂, which can help to passivate the bottom surface of perovskite layer. The C=O group in PMMA can form Lewis-based-Lewis-acid interaction with the uncoordinated Pb of perovskite and hence leads to a reduction of overall trap states (Chem. Eng. J., 2023, 143656; Adv. Energy Mater.2018, 8, 1801208; Energy Environ. Sci., 2017,10, 1792–1800). With the help of such an effect, an over 23% PCE can be achieved for this type of device, which is approaching the state-of-the-art devices with PCE of ~25%. The J-V curves and photovoltaic parameters are shown in Figure S14 and Table S3 in the revised manuscript:

In Supporting Information,

Figure S14. Current density versus voltage curves of optimised pristine and surface-passivated devices with PMMA. Device architecture: ITO/SnO₂/PMMA/FA_{0.99}Cs_{0.01}PbI₃/Spiro-OMeTAD/Au (pristine) and ITO/SnO₂/PMMA/FA_{0.99}Cs_{0.01}PbI₃/OAI/Spiro-OMeTAD/Au (surface-passivated). Detailed photovoltaic parameters are listed in Table S3.

Table S3. Photovoltaic parameters of optimised pristine and surface-passivated devices.

		V _{oc} (V)	J _{sc} (mA cm ⁻²)	FF (%)	PCE (%)
Pristine	Forward	1.04	24.7	64.9	16.7
	Reverse	1.08	24.7	74.3	19.8
Surface-passivated	Forward	1.14	25.1	76.8	21.9
	Reverse	1.15	25.2	80.4	23.2

Then we conduct ns-PPPC measurement to monitor how the trapped carrier behaviour changes after incorporating PMMA in the device. Figures S15a and S15b (also attached below) clearly show the overall n_{TC} in both the pristine and surface-passivated devices with PMMA are smaller than

those without PMMA, which can be attributed to the passivation of the perovskite bottom surface. Moreover, the delayed rise time of the optimised pristine device is still longer than the optimised surface-passivated device (Figure S15c), which is similar to the results we acquired from the original pristine and surface-passivated devices, further consolidating the conclusion we made.

We have added the following discussion and figure in the revised manuscript:

On page 14,

“To investigate how the trapped carrier behaviour changes in higher-performance devices, we optimised the devices via depositing an ultra-thin polymethyl methacrylate (PMMA) layer on top of the SnO₂ layer [i.e., with device architectures of ITO/SnO₂/PMMA/FA_{0.99}CS_{0.01}PbI₃/(OAI)/Spiro-OMeTAD/Au]. PMMA can passivate the bottom surface of perovskite via Lewis-base-Lewis-acid interaction between the C = O group and the uncoordinated Pb, resulting in a reduction of overall trap states.^{56,62,63} Consequently, much improved PCEs from 16.1% to 19.8% for pristine device and from 19.5% to 23.2% for surface-passivated device were achieved (detailed photovoltaic performance can be found in Figure S14 and Table S3). Figures S15a and S15b show the ns-PPPC results for devices with and without PMMA, illustrating that the overall n_{TC} decreases in both pristine and surface-passivated devices after the incorporation of PMMA. This result further confirms the passivation effect of PMMA. Moreover, in the optimised devices, the rise time of pristine device (92.16 ns) is much longer than that in surface-passivated device (16.86 ns) (Figure S15c), which consolidates the photophysical picture developed for the original pristine and surface-passivated devices.”

In the Methods section,

“PMMA solution was prepared by dissolving PMMA (Mw = ~4000) in chlorobenzene with a concentration of 2 mg ml⁻¹.”

“For the device with PMMA layer, the PMMA solution was spin-coated (4000 rpm for 30 s) on top of the SnO₂ layer and followed by annealing at 100 °C for 8 mins in a N₂-filled glovebox.”

In Reference section:

[56] Peng, J. et al. A universal double-side passivation for high open-circuit voltage in perovskite solar cells: role of carbonyl groups in poly(methyl methacrylate). *Adv. Energy Mater.* **8**, 1801208 (2018).

[62] Qiu, W. et al. Low-temperature robust MAPbI₃ perovskite solar cells with power conversion efficiency exceeding 22.4%. *Chem. Eng. J.* **468**, 143656 (2023).

[63] Rui, Y. et al. Defect passivation and electrical conductivity enhancement in perovskite solar cells

using functionalized graphene quantum dots. *Mater. Futures* **1**, 045101 (2022).

In Supporting Information:

Figure S15. Trapped carrier dynamics of optimised pristine and surface-passivated devices [ITO/SnO₂/PMMA/FA_{0.99}Cs_{0.01}PbI₃/(OAI)/Spiro-OMeTAD/Au], under pump fluence of 3.3 $\mu\text{J cm}^{-2}$ and push fluence of 170.8 $\mu\text{J cm}^{-2}$. (a) Pristine and (b) surface-passivated devices with and without PMMA layer. (c) Normalised dynamics of trapped carriers in pristine and surface-passivated devices with PMMA layer. Note that the spike shown in the pristine device is due to the instant optical effect between pump and push pulses, rather than the occurrence of real trapped carrier dynamics.

3. What is missing, is a structure determination to make the reader understand what is the type of perovskite active layer established in the present work? Obviously, there is shortcomings but unclear to some extent, which is the structural origin and then causing the observed trap states.

Reply:

Following the Reviewer's suggestion, we comment in the revised manuscript on the structure of the perovskite layer. Specifically, we address the question whether the deposition of OAI layer could induce the formation of 2D perovskite or not. We measured the X-ray diffraction patterns for pristine and surface-passivated perovskite films and added them as Figures S1e and S1f (as shown below). The results show that only the signal from 3D perovskite is detected, while no fingerprint signal at low diffraction angles ($2\theta < 10^\circ$) from 2D perovskite is observed after the deposition of OAI. Therefore, we consider no 2D perovskite formed on top of the perovskite layer and OAI just act as the surface passivator as discussed in point 1.

To better demonstrate the structure of perovskite, interfacial configuration, and OAI passivation effect at the perovskite surface to readers, we added the following discussion and figures (Figures 1c, S1e and S1f) in our revised manuscript:

On page 5,

“The low-angle ($2\theta < 10^\circ$) X-ray diffraction signal shown in Figures S1e and S1f shows no indication of a 2D (or layered) perovskite phase formed on top of the 3D perovskite surface after the deposition of OAI.”

and,

Figure 1. Device properties of FA_{0.99}Cs_{0.01}PbI₃ PeSCs. (a) Device architecture of the surface-passivated device. The pristine device has a similar architecture but without the OAI layer. (b) Current density versus voltage of the pristine and surface-passivated devices measured during forward and reverse voltage sweeps at a scan rate of 40 mV s⁻¹. (c) Configuration of the surfaces of pristine and surface-passivated perovskites. The OA chains mount to the perovskite surface via their NH₃⁺ groups, which also passivate the FA⁺ vacancies simultaneously, while the I⁻ vacancies can be passivated by the I⁻ in OAI.³⁹

In the Methods section,

“X-ray diffraction (XRD) patterns were obtained from an X-ray diffractometer (Panalytical X’Pert Pro) with an X-ray tube (Cu K α , $\lambda = 1.5406 \text{ \AA}$).”

In Reference section:

[39] Wu, G. *et al.* Surface passivation using 2D perovskites toward efficient and stable perovskite solar cells. *Adv. Mater.* **34**, 2105635 (2022).

In the Supporting Information,

Figure S1. Passivation effect and impact of OAI. X-ray diffraction patterns of (e) pristine and (f) surface-passivated perovskite films. We consider no 2D perovskites formed as only 3D perovskite peaks are observed.

4. Unfortunately, the experimental condition of the solar cell operation are not clear. Was the operation under inert atmosphere or under vacuum? From the used set-up one might suppose vacuum condition. As seen in the literature (Degradation mechanisms of perovskite solar cells under vacuum and one atmosphere of nitrogen, *Nature Energy* 6, 977-986, 2021), such question is highly relevant and needs to be well discussed.

Reply:

We follow the Reviewer’s suggestion to elaborate the information on experimental conditions. We now use the paper cited by the Reviewer as reference 45, as both their and our devices have the same device architecture (ITO/SnO₂/Perovskite/Spiro-OMeTAD/Au) and generate similar device performance (~20% PCE). We also agree that the degradation of PeSCs is dependent on the environmental atmosphere concerning the phase and structural stability.

According to the proposed reference, operating PeSCs under vacuum causes both a large degree of lattice shrinkage and a spontaneous process for phase segregation of the mixed-cation lead mixed-halide PeSCs, while these effects are significantly mitigated in nitrogen. However, in our research, no experiment was conducted under vacuum. As mentioned in point 2, almost all experiments were conducted under ambient conditions, and therefore we need ultra-stable devices that can survive without encapsulation for months. For temperature-dependent measurements, we filled nitrogen gas into the Linkam cryostat chamber, and no vacuum was used.

Therefore, we believe the degradation of perovskite under vacuum conditions is not a factor in our study. We added the environmental condition of our experiments and the following discussion in our revised manuscript:

On page 11,

“The devices were measured in a nitrogen-filled environment to prevent potential degradation of perovskite under vacuum conditions.⁴⁵”

In the Methods section,

“All the devices were used unencapsulated.”

“The experiment was conducted under ambient atmosphere.”

“Nitrogen was fully filled in the chamber to prevent any potential degradation of perovskites.”

In references:

*[45] Guo, R. et al. Degradation mechanisms of perovskite solar cells under vacuum and one atmosphere of nitrogen. *Nat. Energy* **6**, 977–986 (2021).*

5. Charge-carrier trapping and radiative recombination in metal halide perovskite semiconductors as discussed in *Adv. Funct. Mater.* **30**, 2004312 (2020) needs to be taken into better account in the present work.

Reply:

We thank the reviewer for bringing this study to our attention. The results of Herz and coworkers indicate that fast charge trapping occurs in a few nanoseconds due to the rapid localisation of free charge carriers in unoccupied trap states, while trap-assisted recombination occurs on a microsecond timescale. Both values match well with our experimental results and support our interpretation. We have added the following discussion in our revised manuscript:

On page 12,

“Furthermore, the timescale of both bulk trap filling and trap-assisted recombination is consistent with those obtained from trapped-carrier modelling based on the previous report.⁵⁷”

In Reference section,

[57] Trimpl, M. J. *et al.* Charge-carrier trapping and radiative recombination in metal halide perovskite semiconductors. *Adv. Funct. Mater.* **30**, 2004312 (2020).

As a consequence, the present version of the manuscript is not at all suited for publication in any high profile journal and should be moved to a more specialized journal dealing with technical developments and methods.

Reply:

Following the Reviewer’s guidelines and recommendations, we have substantially reworked the manuscript including the new experiments as well as the results on new, more efficient material and device systems. We hope that the revised version will be found of interest for the board readership of Nature Communications.

REVIEWERS' COMMENTS

Reviewer #1 (Remarks to the Author):

The authors use the pump-push CW and transient approach to study the perovskite solar cell devices, particularly, focus on the properties of trap states. The authors did extensive and careful study including the temperature dependence, as suggested by the reviewer.

This is certainly a contribution to the perovskite PV community, and I encourage the editor to consider for a possible acceptance because of the characterization, and careful modelling, and revealing the trapped carrier dynamics in ns time scale and 100s ns time scale due to interface, which is quite valuable to the community.

Further, the author conduct a challenging temperature dependent experiment, which is hard in terms of technical setup.

Reviewer #2 (Remarks to the Author):

The authors have satisfactorily addressed my comments.

Reviewer #3 (Remarks to the Author):

The authors have well responded to my concerns and questions. The additional measurements and discussion have enriched the manuscript substantially and move it to the needed level for high impact work. Thus, I recommend publication in the present form.